# High-dimensional analysis of 16 SARS-CoV-2 vaccine combinations reveals lymphocyte signatures correlating with immunogenicity

Nicolás Gonzalo Nuñez ⬚[1,8,9,10] ✉, Jonas Schmid ⬚[1,10], Laura Power[1,10], Chiara Alberti[1], Sinduya Krishnarajah ⬚[1], Stefanie Kreutmair ⬚[1], Susanne Unger[1], Sebastián Blanco[2], Brenda Konigheim[2], Constanza Marín ⬚[3,4], Luisina Onofrio ⬚[3,4], Jenny Christine Kienzler ⬚[1], Sara Costa-Pereira[1], Florian Ingelfinger[1], InmunoCovidCba*, InViV working group*, Marina E. Pasinovich[5], Juan M. Castelli[5], Carla Vizzotti[5], Maximilian Schaefer ⬚[1], Juan Villar-Vesga[1], Sarah Mundt[1], Carla Helena Merten[1], Aakriti Sethi[1], Tobias Wertheimer[1], Mirjam Lutz[1], Danusia Vanoaica[1], Claudia Sotomayor[3,4], Adriana Gruppi[3,4], Christian Münz ⬚[1], Diego Cardozo[6], Gabriela Barbás[6], Laura Lopez[6], Paula Carreño[6], Gonzalo Castro[6], Elias Raboy[6], Sandra Gallego[2,7,11], Gabriel Morón ⬚[3,4,11], Laura Cervi[3,4,11], Eva V. Acosta Rodriguez ⬚[3,4,11], Belkys A. Maletto[3,4,11], Mariana Maccioni ⬚[3,4,11] ✉ & Burkhard Becher ⬚[1,11] ✉

The range of vaccines developed against severe acute respiratory syndrome coronavirus 2 (SARS-CoV-2) provides a unique opportunity to study immunization across different platforms. In a single-center cohort, we analyzed the humoral and cellular immune compartments following five coronavirus disease 2019 (COVID-19) vaccines spanning three technologies (adenoviral, mRNA and inactivated virus) administered in 16 combinations. For adenoviral and inactivated-virus vaccines, heterologous combinations were generally more immunogenic compared to homologous regimens. The mRNA vaccine as the second dose resulted in the strongest antibody response and induced the highest frequency of spike-binding memory B cells irrespective of the priming vaccine. Priming with the inactivated-virus vaccine increased the SARS-CoV-2-specific T cell response, whereas boosting did not. Distinct immune signatures were elicited by the different vaccine combinations, demonstrating that the immune response is shaped by the type of vaccines applied and the order in which they are delivered. These data provide a framework for improving future vaccine strategies against pathogens and cancer.

The management of the coronavirus disease 2019 (COVID-19) pandemic has relied heavily on the development and global deployment of vaccines that protect against severe acute respiratory syndrome coronavirus 2 (SARS-CoV-2). Numerous homologous prime-boost vaccination regimens were shown to stimulate robust cellular and humoral immune responses and have been approved for clinical use[1]. The many different vaccines and vaccine technologies employed during this pandemic provide a unique opportunity to study their effects

**Fig. 1 | Schematic of vaccination regimens and data analysis pipeline.**
Participants were vaccinated with one of sixteen COVID-19 vaccine combination regimens and donated blood at timepoints T1 (4–12 weeks after dose 1), T2 (2 weeks after dose 2) and T3 (4 weeks after dose 2). PBMCs collected at T1 and T3 were used for IFNγ ELISpot assays and high-dimensional spectral flow cytometry analysis. Sera and plasma collected at T1, T2 and T3 were analyzed for anti-S-RBD IgG levels and NAb titers, respectively. Anti-S-RBD IgA levels were measured at T3. Participants were monitored for adverse events. The data collected were analyzed using dimensionality reduction, FlowSOM-based clustering algorithms and statistical testing. IFNγ, interferon-gamma.

and dissect the immune responses associated with each regimen. Different reports have shown that certain heterologous prime-boost regimens provide enhanced immunogenicity against SARS-CoV-2 (refs. 2–5; https://www.who.int/publications/i/item/WHO-2019-nCoV-vaccines-SAGE-recommendation-heterologous-schedules), in line with preclinical data comparing heterologous versus homologous vaccination regimens against numerous other pathogens[6–11]. Nevertheless, little is known about specific differences in the cellular immune response to different types of vaccines and their mix-and-match combinations as well as the cellular phenotypes that are associated with potent immunization, not only in the context of SARS-CoV-2 but also other infectious diseases. The variety of vaccines against COVID-19 offers a unique opportunity to characterize the human immune response to a range of vaccine technologies.

The primary COVID-19 vaccination regimens approved for clinical use include viral-vector-based approaches such as AZD1222 (two-dose homologous prime-boost, henceforth termed AZD), developed by the University of Oxford in collaboration with AstraZeneca[12]; Sputnik V (prime-boost heterologous human adenoviral vectors, serotypes 26 (henceforth Sput-26) and 5 (Sput-5)) by the Gamaleya Research Institute[1,13] and Ad5-nCoV-S (a single-dose viral vector vaccine, henceforth Ad5) by CanSino Biologics[14]. Moreover, inactivated vaccines are also approved such as BBIBP-CorV (henceforth BBIBP), developed by Sinopharm[15], and mRNA-based vaccines such as mRNA-1273 by Moderna[16], as well as protein subunit vaccines such as NVX-CoV2373, a recombinant spike protein nanoparticle vaccine by Novavax[17] (the latter was not part of this study). These vaccines are being widely administered on a global scale (https://covid19.trackvaccines.org/agency/who/).

In 2021, the Argentine Ministry of Public Health began a multi-center study (ECEHeVac, NCT04988048) to systematically compare the immune responses to homologous and heterologous COVID-19 vaccination regimens in Argentina[18]. In the present study, we specifically analyzed samples from one cohort located at a clinical center in the city of Córdoba (REPIS-Cba 4371). The individuals recruited for the trial were first vaccinated with one dose of AZD, BBIBP, Sput-26 or mRNA-1273, followed by a randomized second dose of AZD, BBIBP, Sput-26, Sput-5, mRNA-1273 or Ad5, resulting in a total of 16 different homologous or heterologous prime-boost regimens. We conducted a head-to-head comparison of the immune responses to these COVID-19 vaccine regimens and interrogated antibody and cellular responses, which are known to act in tandem to provide protection against SARS-CoV-2 (ref. 19).

Through the application of high-dimensional single-cell analysis, we identified a stronger IgM expression among spike-binding memory B cells, a higher frequency of CD4−CD8− T cell clusters as well as a strong spike-specific T cell response in vaccine combinations that included priming with the inactivated virus vaccine BBIBP. Additionally, regimens that induced spike-binding memory B cells skewed to a switched-activated (CD21−CD38−/lo) phenotype and a downregulation of CXCR5 were associated with higher antibody titers and stronger serum-neutralizing activity. Cells with a comparable expression profile have previously been described to transiently arise after influenza vaccination and to be potential precursors of long-lived plasma cells[20,21]. Our study paves the way to further our understanding of the immune response to different vaccine strategies and supports the rationale for combining different technologies for future development of improved vaccines against pathogens and cancer.

## Results
### Study cohort and reactogenicity
To compare the immune profiles resulting from different COVID-19 vaccination regimens, we enrolled participants to be vaccinated with the regimens listed in Fig. 1. Individuals were given a priming dose (dose 1) followed by a homologous or heterologous booster dose (dose 2) 4–12 weeks later, resulting in a total of 16 combinatorial vaccination regimens. The interval between dose 1 and dose 2 depended on the first vaccine administered: 4 weeks for individuals primed with mRNA-1273 or BBIBP, 9–10 weeks after priming with AZD or 12 weeks after priming with Sput-26. Blood samples were collected from each volunteer at the day of dose 2 administration (T1), 14 ± 2 d after dose 2 (T2) and 28 ± 1 d after dose 2 (T3; Table 1). The T1–T2 and T2–T3 intervals were approximately 2 weeks for all groups (Table 1). Serum and plasma were collected at all timepoints, and peripheral blood mononuclear cells (PBMCs) were isolated at T1 and T3. In total, 1,491 samples from 497 individuals were included in the analysis. Characteristics of the whole cohort and the vaccine groups are summarized in Table 1. Individuals with a prior history of COVID-19 or a positive SARS-CoV-2 nucleocapsid protein IgG ELISA result at T1 were excluded. The vaccine groups were comparable in number of participants and sex distribution.

All study participants were monitored for up to 6 months after dose 2 to assess reactogenicity. Pain at the site of injection was the most frequent local adverse event after dose 1 (Extended Data Fig. 1a) and dose 2 (Extended Data Fig. 1b). There was no systematic clinical follow up of the cohort in terms of SARS-CoV-2 infections. Until July 2022,

**Table 1 | Cohort characteristics. Age, gender and dose intervals for each vaccine combination**

| Dose 1 | | AZD | | | | | BBIBP | | | | mRNA-1273 | | | | Sput-26 | | |
|---|---|---|---|---|---|---|---|---|---|---|---|---|---|---|---|---|---|
| Dose 2 | Overall, n=497 | Ad5, n=20 | AZD, n=41 | BBIBP, n=25 | mRNA-1273, n=32 | Sput-26, n=29 | AZD, n=25 | BBIBP, n=40 | mRNA-1273, n=30 | Sput-26, n=33 | mRNA-1273, n=20 | Ad5, n=29 | AZD, n=36 | BBIBP, n=34 | mRNA-1273, n=42 | Sput-26, n=25 | Sput-5, n=36 |
| Age | 41.8±15.6 | 32.6±9.4 | 50.9±12.9 | 39.1±10.4 | 43.4±10.9 | 46.2±11.5 | 26.2±13.5 | 29.1±13.0 | 26.8±11.8 | 32.4±14.2 | 26.9±10.3 | 39.2±14.8 | 53.2±11.0 | 52.1±12.3 | 54.7±9.2 | 47.7±10.8 | 49.5±13.1 |
| Gender F | 252(51%) | 10(50%) | 21(51%) | 12(48%) | 14(44%) | 14(48%) | 13(52%) | 19(48%) | 16(53%) | 18(55%) | 11(55%) | 20(69%) | 16(44%) | 13(38%) | 23(55%) | 12(48%) | 20(56%) |
| Gender M | 245(49%) | 10(50%) | 20(49%) | 13(52%) | 18(56%) | 15(52%) | 12(48%) | 21(52%) | 14(47%) | 15(45%) | 9(45%) | 9(31%) | 20(56%) | 21(62%) | 19(45%) | 13(52%) | 16(44%) |
| T1 (days after dose 1) | 64.8±23.7 | 73.6±9.6 | 65.7±6.9 | 65.2±3.1 | 66.9±16.9 | 62.4±7.7 | 33.3±3.1 | 45.8±17.6 | 35.1±4.7 | 38.9±12.9 | 32.9±2.8 | 79.0±12.7 | 82.9±18.9 | 85.0±27.0 | 84.8±17.3 | 82.4±8.0 | 83.6±18.4 |
| T2 (days after dose 2) | 14.3±1.5 (n=437) | 14.2±0.5 (n=18) | 14.2±0.6 (n=35) | 14.3±0.7 (n=23) | 14.5±2.6 (n=29) | 14.1±0.4 (n=28) | 14.4±0.8 (n=22) | 14.8±1.8 (n=26) | 14.2±0.7 (n=27) | 15.1±2.7 (n=27) | 14.1±0.3 (n=14) | 14.3±0.7 (n=27) | 14.7±3.3 (n=33) | 14.1±0.6 (n=34) | 14.0±0.0 (n=39) | 14.2±0.7 (n=23) | 14.1±0.6 (n=32) |
| T3 (days after dose 2) | 28.4±1.2 (n=420) | 29.3±0.9 (n=19) | 28.6±1.6 (n=30) | 28.4±0.9 (n=20) | 28.1±0.8 (n=27) | 28.2±0.9 (n=27) | 28.1±0.3 (n=22) | 28.8±1.6 (n=26) | 28.0±0.0 (n=23) | 28.2±1.4 (n=26) | 28.7±1.4 (n=13) | 29.2±1.6 (n=26) | 28.2±0.9 (n=31) | 28.5±1.7 (n=34) | 28.1±0.3 (n=38) | 28.0±0.0 (n=23) | 28.2±1.1 (n=35) |

Mean ±s.d. number of days and standard deviation after dose 1 or dose 2 were calculated from a subset of the total participants, as indicated below each mean.

however, no deaths were reported among the volunteers. The most common systemic adverse manifestations were fever, arthromyalgia and headache. In general, the spectrum of reported adverse events was comparable across all groups. Of note, the use of mRNA-1273 or Sput-26 as dose 2 in heterologous regimens, and the combination of AZD/Ad5 (dose 1/dose 2), increased the frequency of reported cumulative adverse events compared to homologous regimens. All reactogenicity symptoms were short lived, and there were no hospitalizations due to adverse events. Thus, all vaccine regimens were considered collectively well-tolerated.

### Strength of antibody response varies between vaccines

We evaluated the antibody response to the 16 prime-boost regimens by measuring participants' serum levels of IgG specific for the receptor-binding domain (RBD) of the SARS-CoV-2 spike protein (anti-S-RBD IgG) and their SARS-CoV-2-neutralizing antibody (NAb) titers at timepoints T1, T2 and T3 for all vaccination groups, as well as anti-S-RBD IgA serum levels at T3 (Fig. 2a–e). All study participants had detectable anti-S-RBD IgG levels (>50 AU ml$^{-1}$) after dose 2, and anti-S-RBD IgG levels increased substantially from T1 to T3 in all groups except for AZD/BBIBP (Fig. 2a). At T3, anti-S-RBD IgG and IgA levels as well as NAb titers were highest in individuals that received homologous mRNA-1273/mRNA-1273 vaccination, followed by the heterologous groups that included mRNA-1273 as dose 2 and BBIBP/Sput-26 for IgA (Fig. 2c–e). While in the homologous mRNA-1273/mRNA-1273 group the fold change from the first to the second dose was moderate in comparison (7.4-fold for anti-S-RBD IgG and 19.3-fold for NAb) due to already higher values after the first dose, both of these humoral responses were most strongly enhanced by mRNA-1273 in the heterologous combinations (AZD/mRNA-1273: 51.4-fold and 33.5-fold; BBIBP/mRNA-1273: 70-fold and 72.2-fold; Sput-26/mRNA-1273: 33.1-fold and 41.5-fold), except after priming with AZD, where boosting with Ad5 led to the greatest increase (41.1-fold) in NAb titers (Extended Data Fig. 2a,b). Anti-S-RBD IgG levels were positively correlated with NAb titers at T3 (except for AZD/Ad5; Extended Data Fig. 2c). Thus, heterologous vaccine combinations generally elicited enhanced antibody responses compared to homologous combinations, except when BBIBP was administered as dose 2, and mRNA-1273 as dose 2 induced the strongest antibody responses, regardless of the vaccine used as dose 1.

### mRNA booster potently expands spike-binding memory B cells

B cells contribute to immunological memory against many viral infections and are important in protecting against COVID-19 (ref. 22). The ability of COVID-19 vaccines to induce memory B cell (mBC) expansion is therefore critical for their effectiveness. Hence, we generated a multiparametric flow cytometry panel to interrogate postvaccination B cell dynamics, using fluorescently labeled wild-type SARS-CoV-2 spike multimer probes[23] to identify mBCs specific for the SARS-CoV-2 spike protein. Data were projected using Uniform Manifold Approximation and Projection (UMAP) in conjunction with FlowSOM clustering to evaluate the canonical B cell populations and identify spike-binding mBCs (Fig. 3a and Extended Data Fig. 3a). The proportions of all canonical B cell populations were similar among the 16 vaccinated groups at T3 (Fig. 3b). This suggests that general vaccination strategies do not impact canonical B cell frequencies in the first weeks postvaccination, regardless of the vaccine strategy used. One month after dose 2 (T3), we observed a significant increase in the frequency of the specific spike-binding mBCs in all vaccine groups, except for those with BBIBP as dose 2 (Fig. 3c). The most pronounced increase in spike-binding mBCs and the highest frequencies at T3 were observed in individuals who received mRNA-1273 as dose 2 (Fig. 3c,d and Extended Data Fig. 3b). We found that the frequency of spike-binding mBCs correlated with the levels of anti-S-RBD IgG, anti-S-RBD IgA and NAbs for some (but not all) vaccine groups (Fig. 3e). We did not observe a correlation

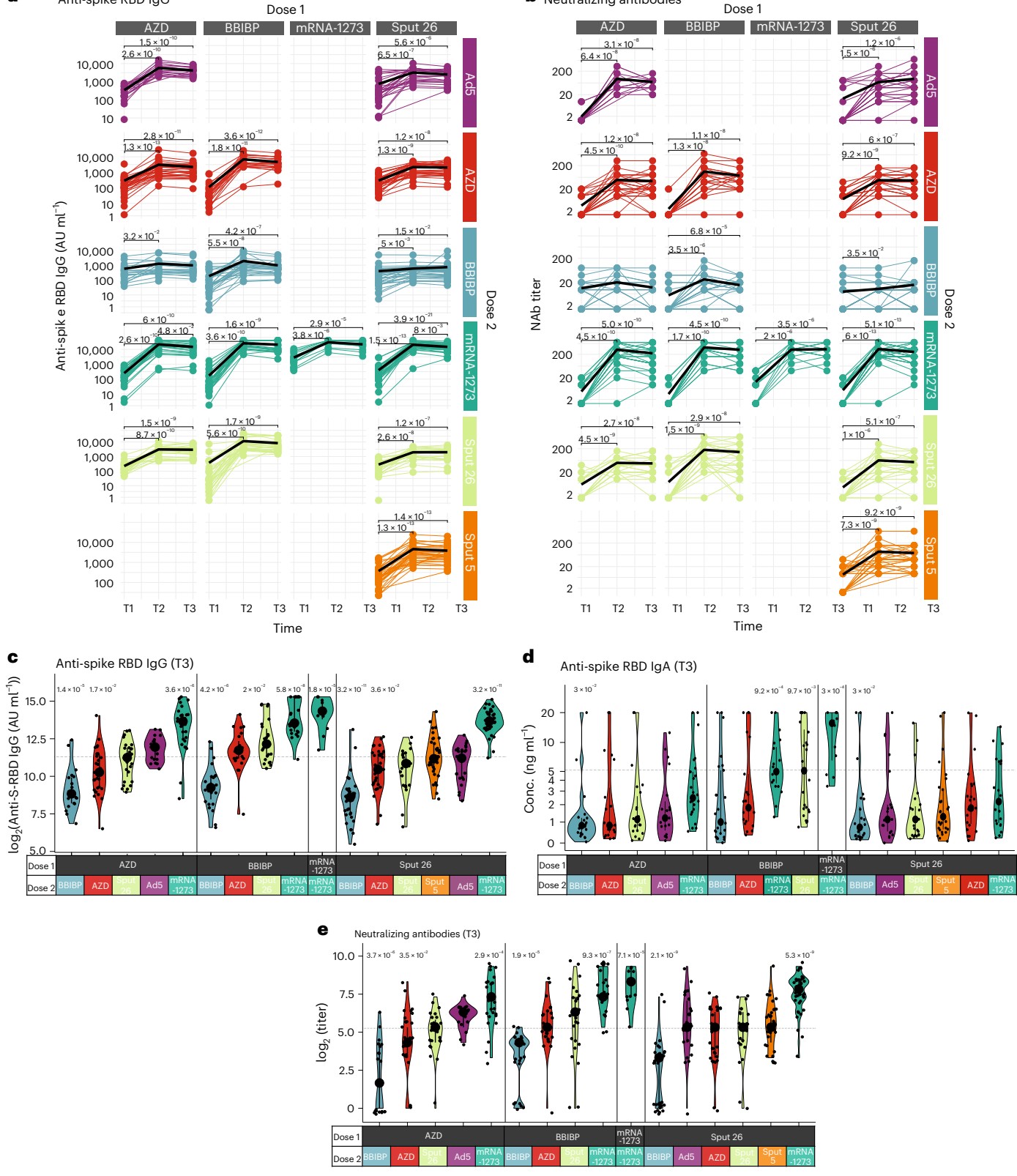

**Fig. 2 | Antibody response in participants after vaccination with various regimens. a,b,** Longitudinal (**a**) anti-S-RBD IgG levels (*n* = 1354) and (**b**) NAb titers (*n* = 1355) measured at timepoints T1, T2 and T3. Black lines show the median. **c–e,** Anti-S-RBD IgG levels (*n* = 420) (**c**), anti-S-RBD IgA levels (*n* = 349) (**d**) and NAb titers (*n* = 421) at T3 (**e**). Large black dots depict the median of each group, and the vertical line spans the interquartile range. The horizontal line shows the overall mean of all participants, and *P* values indicate differences between the respective group and the overall mean of all participants (**c–e**). *P* values were calculated using the Mann–Whitney–Wilcoxon test and corrected for multiple hypothesis testing with the Benjamini–Hochberg method. Only statistically significant *P* values (*P* < 0.05) are shown.

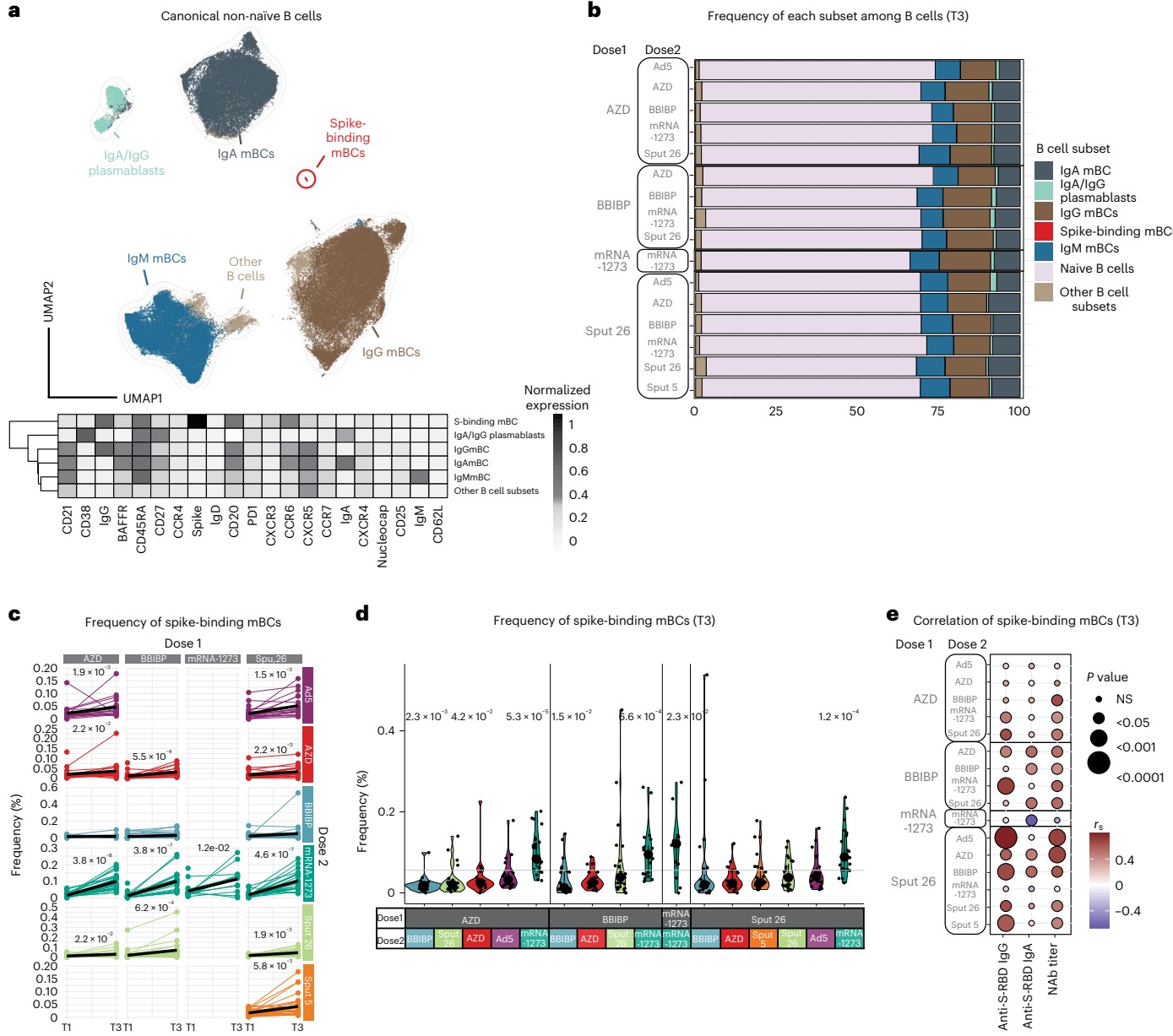

**Fig. 3 | Characterization of spike-binding mBCs from participants receiving different vaccine regimens. a**, UMAP showing the FlowSOM-guided manual metaclustering of nonnaive B cells (IgD⁻/IgM⁻) for all vaccine groups (*n* = 799). The heatmap indicates the median intensity of normalized marker expression (range: 0–1) for the identified B cell subsets. **b**, Relative frequencies of B cell subsets from each vaccine regimen at T3 (*n* = 347). **c**, Longitudinal analysis of spike-binding mBC frequencies among total B cells at timepoints T1 and T3. Black lines show the median (*n* = 754). **d**, Frequencies of spike-binding mBCs among total B cells at T3. Large black dots show the median of each group, and the vertical line spans the interquartile range. The horizontal line indicates the overall mean of all participants. *P* values (shown above groups) indicate differences between the respective group and the overall mean of all participants (*n* = 347). **e**, Correlations of spike-binding mBC frequencies with anti-S-RBD IgG levels, anti-S-RBD IgA levels and NAb titers for each vaccine regimen at T3 (*n* = 347). *P* values were calculated using the Mann–Whitney–Wilcoxon test and corrected for multiple hypothesis testing with the Benjamini–Hochberg method (**c** and **d**). Only statistically significant *P* values (*P* < 0.05) are displayed. Color indicates Spearman's rank correlation coefficient (*r*ₛ), and the bubble size indicates the *P* value.

between other B cell subsets and the antibody response in most of the vaccine groups (Extended Data Fig. 3c).

**Vaccine regimens elicit distinct memory B cell phenotypes**
To better understand the effect of various vaccine regimens on the phenotype of spike-binding mBCs, we evaluated the expression of differentiation markers and trafficking molecules on these cells at T3 (Fig. 4a). Spike-binding mBCs from mRNA-1273-boosted individuals showed a low expression of CD21, CD38 and CXCR5, while these markers were highly expressed in BBIBP-boosted individuals. Furthermore,

spike-binding mBCs from BBIBP-primed groups showed the lowest relative expression of IgG and the highest of IgM. The highest intensity of IgA was found in spike-binding mBCs from individuals vaccinated with mRNA-1273/mRNA-1273 (Fig. 4a), which also resulted in the highest concentrations of anti-SARS-CoV-2 S-RBD IgA in the sera of participants (Fig. 2d). Next, we examined whether these phenotypes were indicative of the strength of the antibody response (Fig. 4b,c and Extended Data Fig. 3d). We found strong negative correlations between antibody response (anti-S-RBD IgG levels and NAb titers) and the expression of CD21, CD38 and CXCR5 on spike-specific mBCs, while IgA and CXCR3

were positively correlated with antibody response (Fig. 4b). Thus, not only the frequency but also the phenotype of antigen-specific mBCs can reflect the strength of the antibody response after COVID-19 vaccination (Fig. 4c). To assess the applicability of these findings, we divided the cohort into positive and negative responders for NAbs (positive responders were defined as NAb titer ≥ 10). Positive responders showed a lower expression of CD21, CD38 and CXCR5 on spike-binding mBCs, thus corroborating their role as B cell markers for a potent antibody response (Fig. 4d and Extended Data Fig. 3e). Additionally, we found that Ad5, mRNA-1273 and Sput-5 led to a positive response for NAbs in 100% of individuals receiving these vaccines as dose 2 (Fig. 4e), whereas the other regimens did not induce detectable NAb titers in all individuals, that is AZD/AZD (no detectable NAb in 2/20 participants), Sput-26/AZD (1/23), AZD/BBIBP (10/20), BBIBP/BBIBP (5/24), Sput-26/BBIBP (9/23), AZD/Sput-26 (1/22), BBIBP/Sput-26 (2/24) and Sput-26/Sput-26 (2/23; Fig. 4e). Equally, none of the booster vaccines converted 100% of individuals to anti-S-RBD IgA positivity (Fig. 4f). We did not find any differences between positive and negative anti-S-RBD IgA responders (cut-off: 0.32 ng ml$^{-1}$) for the spike-binding mBC markers described above (Extended Data Fig. 3f).

Overall, our results highlight two key points. First, boosting with BBIBP resulted in mBCs compatible with a switched-resting (CD21$^+$CD27$^+$CD38$^{+/low}$) or preswitched (IgM$^+$CD21$^+$CD27$^+$CD38$^{+/low}$) phenotype and induced the lowest antibody titers and neutralizing activity in the cohort. Boosting with mRNA-1273, meanwhile, induced phenotypes indicating switched-activated (IgG/IgA$^+$CD21$^-$CD27$^+$CD38$^-$) or atypical (IgG/IgA$^+$CD21$^-$CD27$^-$CD38$^{low}$) mBCs[24] and led to the strongest antibody response. Thus, the B cell immune landscape in vaccinated individuals clearly mirrors the antibody profile seen in the serum shortly after vaccination. Second, spike-binding mBCs from BBIBP-primed individuals showed expression profiles suggesting different isotype switching (indicated by higher IgM expression), representing a characteristic memory B cell signature that persists after dose 2.

## BBIBP priming enhances SARS-CoV-2-specific T cell response

Increased frequencies of interferon-gamma (IFNγ)-secreting T cells against SARS-CoV-2 spike, nucleoprotein and matrix proteins are known to predict protection after vaccination from COVID-19 (refs. 1,25). To assess the cellular immune response upon antigen re-encounter, we stimulated PBMCs from timepoints T1 and T3 with SARS-CoV-2 spike and nucleocapsid peptide pools and measured IFNγ production with an ELISpot assay (Fig. 5a–c). BBIBP-primed individuals showed significant increases in spike-induced IFN-γ production with all booster vaccines. For Sput-26-primed individuals, only Sput-5 (2.7-fold) as dose 2 resulted in a significant increase of spike-specific responses at T3 (Fig. 5a and Extended Data Fig. 4a). At T3, the strongest response in AZD-primed individuals was observed in the AZD/Ad5 group, while BBIBP-primed individuals reached high levels with all vaccine combinations (Fig. 5c). Meanwhile, the nucleocapsid-induced IFNγ production at T3 in BBIBP-primed participants was highest after boosting with mRNA-1273, BBIBP or AZD (Extended Data Fig. 4b).

Overall, BBIBP-primed individuals showed the highest spike-induced IFNγ production, independent of dose 2. BBIBP is the only vaccine in the study that targets the whole SARS-CoV-2 virus rather than solely the spike protein[15]. Therefore, as expected, this was the only vaccine that induced median IFNγ production levels above the cut-off threshold when stimulated with nucleocapsid peptides (Extended Data Fig. 4b). Interestingly, we did not detect responses above the threshold level against both antigens when BBIBP was given as dose 2 in heterologous regimens. However, we also observed nucleocapsid-specific IFNγ production in a small proportion of participants vaccinated with vaccines only targeting the spike protein, potentially reflecting previously reported cross-reactive T cell immunity against SARS-CoV-2 nucleocapsid in individuals without prior SARS-CoV-2 infection or vaccination[26]. In sum, these results reveal that T-cell-mediated responses against SARS-CoV-2 spike and nucleocapsid peptides are stronger when BBIBP is administered as dose 1, regardless of the vaccine used as dose 2.

## T cell phenotypes correlate with spike-specific IFNγ response

Given the differences in the T cell responses, we interrogated overall T cell dynamics to determine whether the frequencies of some T cell subsets are indicative of strong spike-specific cellular responses upon antigen re-encounter. We began by generating a lymphocyte-focused panel for single-cell analysis and defined 15 canonical T cell subpopulations using naive/memory-associated markers (Extended Data Fig. 5a–c). No significant differences were observed in the relative frequencies of these subsets among the vaccinated groups at T3 (Extended Data Fig. 5d), and none of the subsets showed a strong correlation ($|r_s| > 0.25$ and $P < 0.05$) with spike-induced IFNγ production (Extended Data Fig. 5e). To investigate the T cell subsets more deeply, we reclustered the samples using 27 functional and lineage-specific spectral flow parameters (Extended Data Fig. 5a) and performed FlowSOM clustering on the T cell compartment. The frequencies of four of the resulting T cell clusters correlated with the levels of spike-induced IFNγ production detected in the ELISpot assay ($|r_s| > 0.25$ and $P < 0.05$; Supplementary Table 1, Fig. 5d–e and Extended Data Fig. 5f). The phenotypes of the correlating clusters are depicted in Fig. 6a. The clusters that positively correlated (CM CD4$^+$ cluster 9, EM CD8$^+$ clusters 71&72 and CD4$^-$CD8$^-$ clusters 3&8), respectively, expressed high levels of CD38, KLRG1 and CD27 compared to the canonical T cell populations (Fig. 6b). Next, we compared the frequency of these T cell clusters across the 16 vaccine regimens, finding that BBIBP-primed combinations showed higher frequencies of CD4$^-$CD8$^-$ clusters 3&8 compared to the other groups (Fig. 5f). We then investigated differences between positive (≥1.03 normalized ELISpot response) and negative (<1.03) responders independent of the vaccine group. In general, positive responders were significantly higher for the described positively correlating T cell clusters, higher anti-S-RBD IgG and NAb titers as well as a lower expression of CD21 and CD38 on spike-binding mBCs (Fig. 6c and Extended Data Fig. 5g). The proportion of positive responders at T3 was highest after priming with BBIBP (Fig. 6d). In sum, we identified postvaccination T cell clusters that are markers for an antigen-specific T cell response upon antigen re-encounter and are differentially expressed in BBIBP-primed individuals.

**Fig. 4 | Phenotypes of spike-binding mBCs from participants receiving different vaccine regimens. a**, Scaled expression of phenotypic markers by spike-binding mBCs at T3 for each group. *P* values indicate differences between the respective group and the overall mean of all participants for each marker (*n* = 347). **P* < 0.05, ***P* < 0.01, ****P* < 0.001. **b**, Correlations between phenotypic marker expression by spike-binding mBCs and antibody response (anti-S-RBD IgG levels, anti-S-RBD IgA levels and NAb titers) across all vaccine regimens at T3. Color indicates Spearman's rank correlation coefficient (rs), and the bubble size indicates the *P* value (*n* = 347). **c**, Spearman's rank correlations between CD21, CD38 and CXCR5 expression and anti-S-RBD antibodies and NAb titers (*n* = 339). **d**, Participants were classified as negative (<10) or positive (≥10) for NAb titers and mean antibody levels. Spike-binding mBC marker expression was compared between negative and positive participants (*n* = 347). Boxes bound the IQR divided by the median, and Tukey-style whiskers extend to a maximum of 1.5× IQR beyond the box. Dots are participant data points. *P* values were calculated using the Mann–Whitney–Wilcoxon test and corrected for multiple hypothesis testing with the Benjamini–Hochberg method (**a** and **d**). Only statistically significant *P* values (*P* < 0.05) are displayed. **e**, The count and percentage of participants at T1 and T3 classified as negative or positive responders, grouped by vaccine regimen (*n* = 407 at T1, *n* = 347 at T3). **f**, Counts and percentages of participants negative or positive for anti-S RBD IgA at T3 per vaccine regimen (*n* = 349). IQR, interquartile range.

**Immune signatures segregate four major vaccination groups**

Side-by-side analysis of our humoral and cellular immunogenicity data (Figs. 2c–e, 3d and 5c) revealed that BBIBP/mRNA-1273 induced especially high responses in both immune compartments (Fig. 7a). Only for Sput-26/Ad5 did antibody responses correlate with T cell responses after COVID-19 vaccination (Extended Data Fig. 6a).

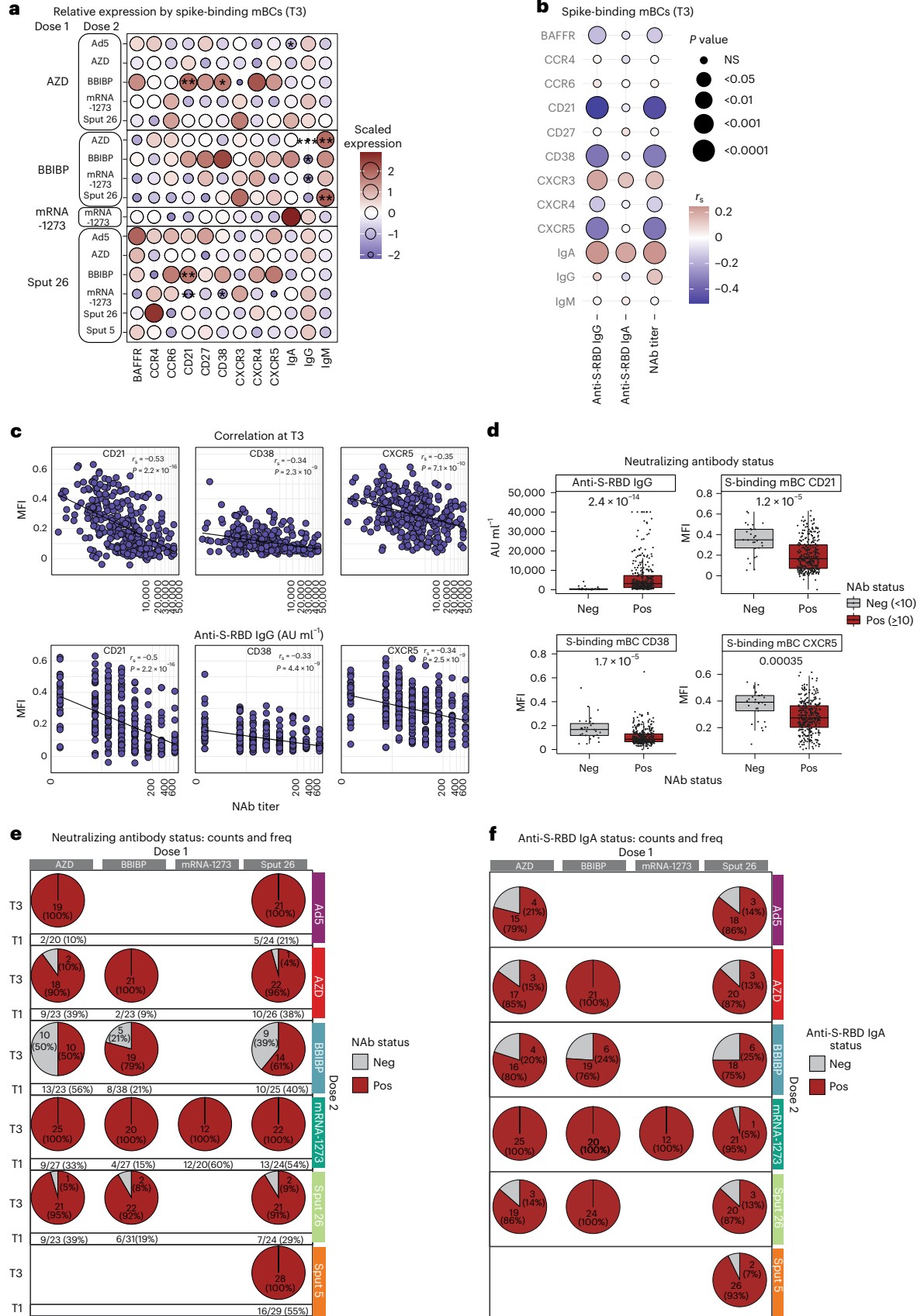

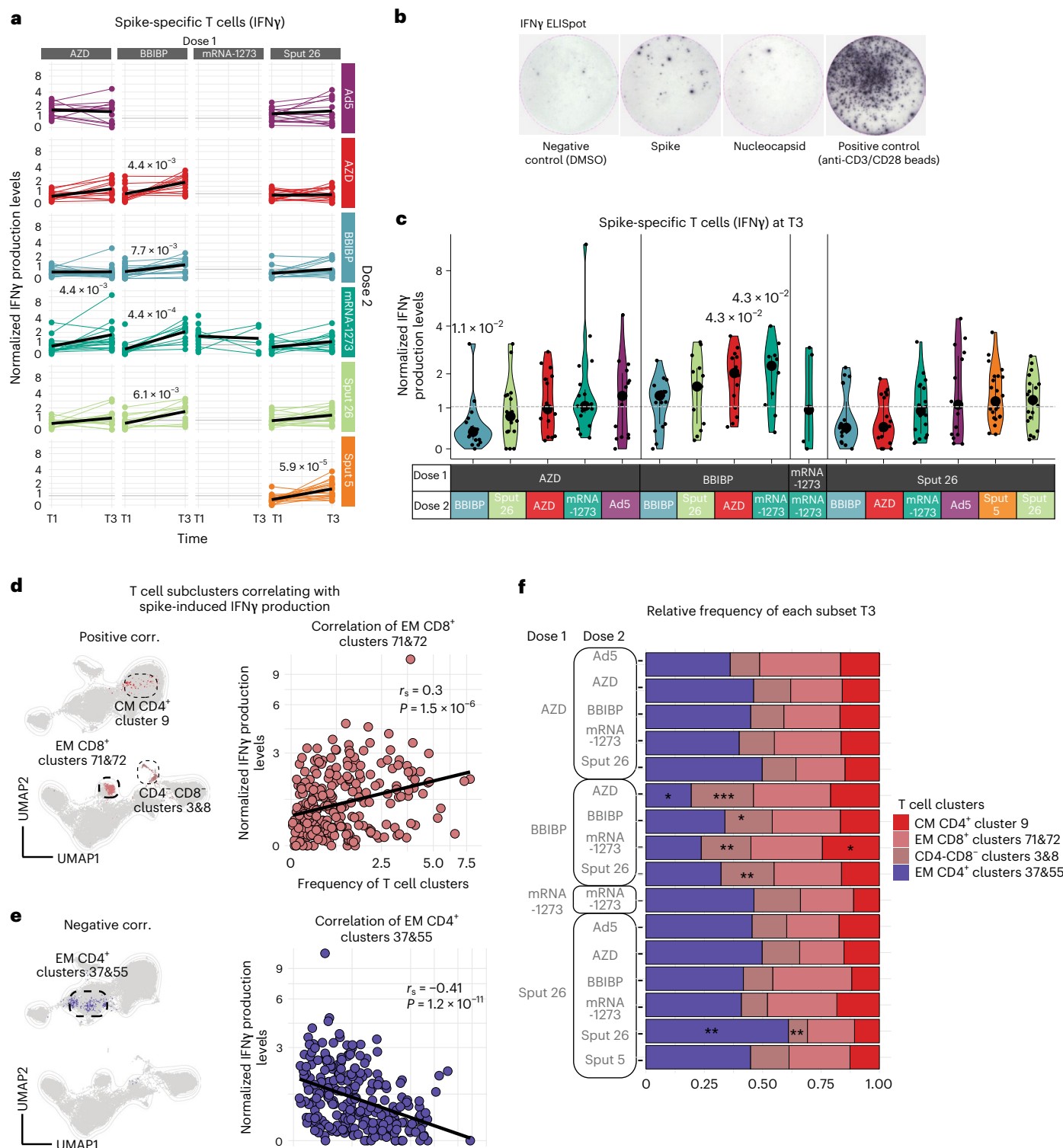

**Fig. 5 | Spike-specific T cell responses to antigen re-encounter after delivery of various vaccine regimens. a,b**, Normalized IFNγ production after stimulation of PBMCs with SARS-CoV-2 spike peptide pool measured by ELISpot assay. Longitudinal IFNγ response at T1 and T3. Black lines show the median ($n = 581$) (**a**). Representative images of one IFNγ ELISpot assay (**b**). **c**, Normalized IFNγ responses at T3 after stimulation with SARS-CoV-2 spike peptide pool ($n = 255$). Large black dots show the median of each group, and the vertical line spans the IQR. The horizontal line indicates the positivity threshold. $P$ values indicate differences between the respective group and the overall mean of all participants. $P$ values were calculated using the Mann–Whitney–Wilcoxon test and the Benjamini–Hochberg method to control for multiple hypothesis testing; only statistically significant $P$ values ($P < 0.05$) are shown (**a** and **c**).

**d,e**, UMAP of the T cell compartment (CD3⁺) showing the clusters with frequencies (**d**) positively and (**e**) negatively correlated with SARS-CoV-2 spike peptide-induced IFNγ response. Scatter plots show the frequencies of the clusters with the (**d**) highest and (**e**) lowest Spearman's rank correlation coefficients ($r_s$) when compared to IFNγ production ($n = 255$). **f**, Relative frequency per vaccine combination of the T cell subclusters with the highest positive or negative correlations to the SARS-CoV-2 spike peptide-induced IFNγ levels ($n = 347$). $P$ values indicate differences between the specific group and the overall mean frequency of all participants of each T cell cluster per vaccine combination and were calculated using the Mann–Whitney–Wilcoxon test and the Benjamini–Hochberg method to control for multiple hypothesis testing; *$P < 0.05$, **$P < 0.01$ and ***$P < 0.001$. Only statistically significant $P$ values ($P < 0.05$) are shown.

**a**

Phenotype of main T cell subsets vs T cell clusters correlating with S-induced IFNγ

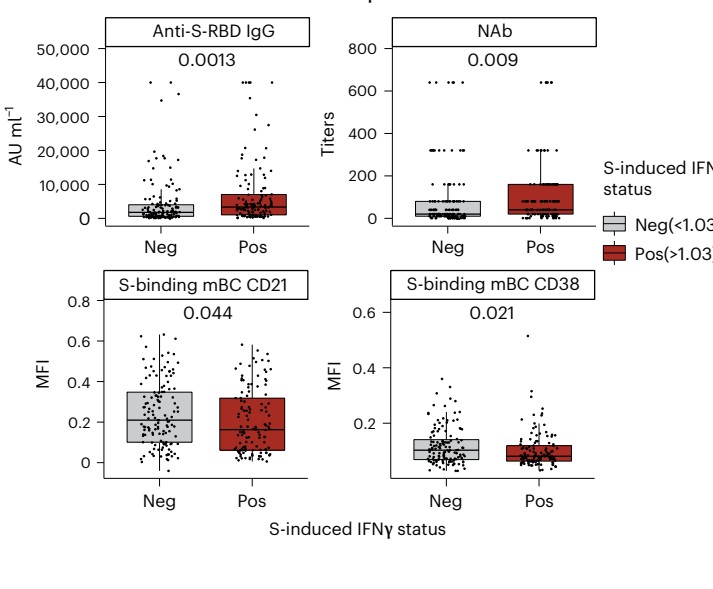

**b**

Differential expression between selected and canonical T cell clusters

**c**

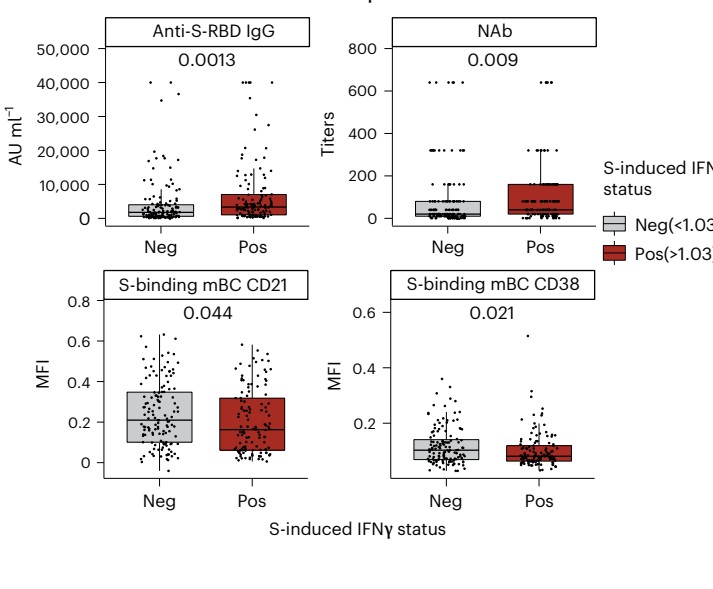

**d**

Spike-induced IFNγ status: counts and freq

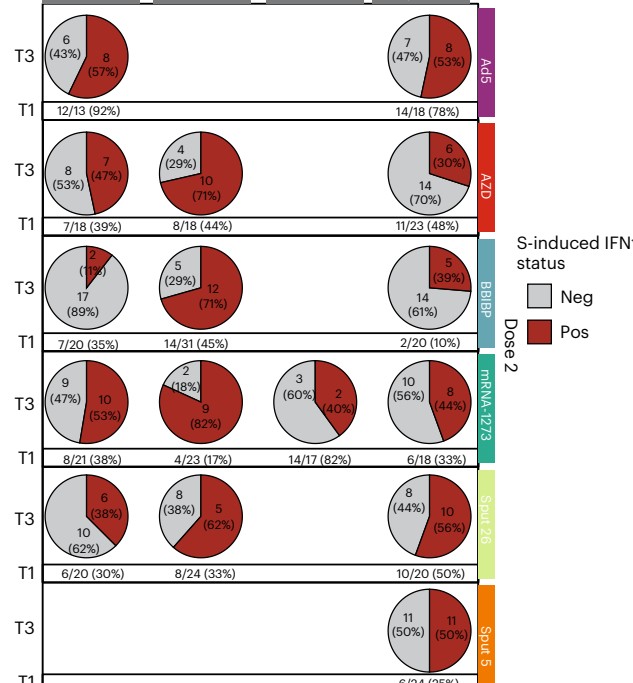

**Fig. 6 | Phenotypes of the T cell subclusters with the highest positive and negative correlation to SARS-CoV-2 spike peptide-induced IFNγ production.** **a**, Heatmap showing the median intensity of normalized marker expression (range: 0–1) for canonical T cell subsets and identified T cell subclusters (*n* = 347). **b**, Differential marker expression by the specific T cell subclusters compared to the canonical T cell subsets that positively and negatively correlated with SARS-CoV-2 spike peptide-induced IFNγ responses, filtered for markers with at least 0.2 differential expression. Color and bubble size indicate the differential expression values compared to the canonical cell type (*n* = 347). **c**, Participants were classified as negative (<1.03, the detection threshold) or positive (>1.03) for IFNγ response and mean antibody levels and spike-binding mBC marker expression levels were compared between negative and positive participants. Boxes bound the IQR divided by the median, and Tukey-style whiskers extend to a maximum of 1.5× IQR beyond the box. Dots are participant data points (*n* = 255). **d**, The count and percentage of participants classified as negative or positive responders, grouped by vaccine regimen at T1 and T3 (*n* = 328 at T1, *n* = 255 at T3).

We then combined the mBC response markers from Fig. 4b,c and the T cell clusters shown in Fig. 6b to better characterize the specific cellular immune profiles associated with the 16 vaccine regimens.

K-means clustering resulted in the segregation of four main vaccine regimen signatures (that is, boost BBIBP, viral vector, boost mRNA and prime BBIBP) clearly separated through the comparison of 11 parameters

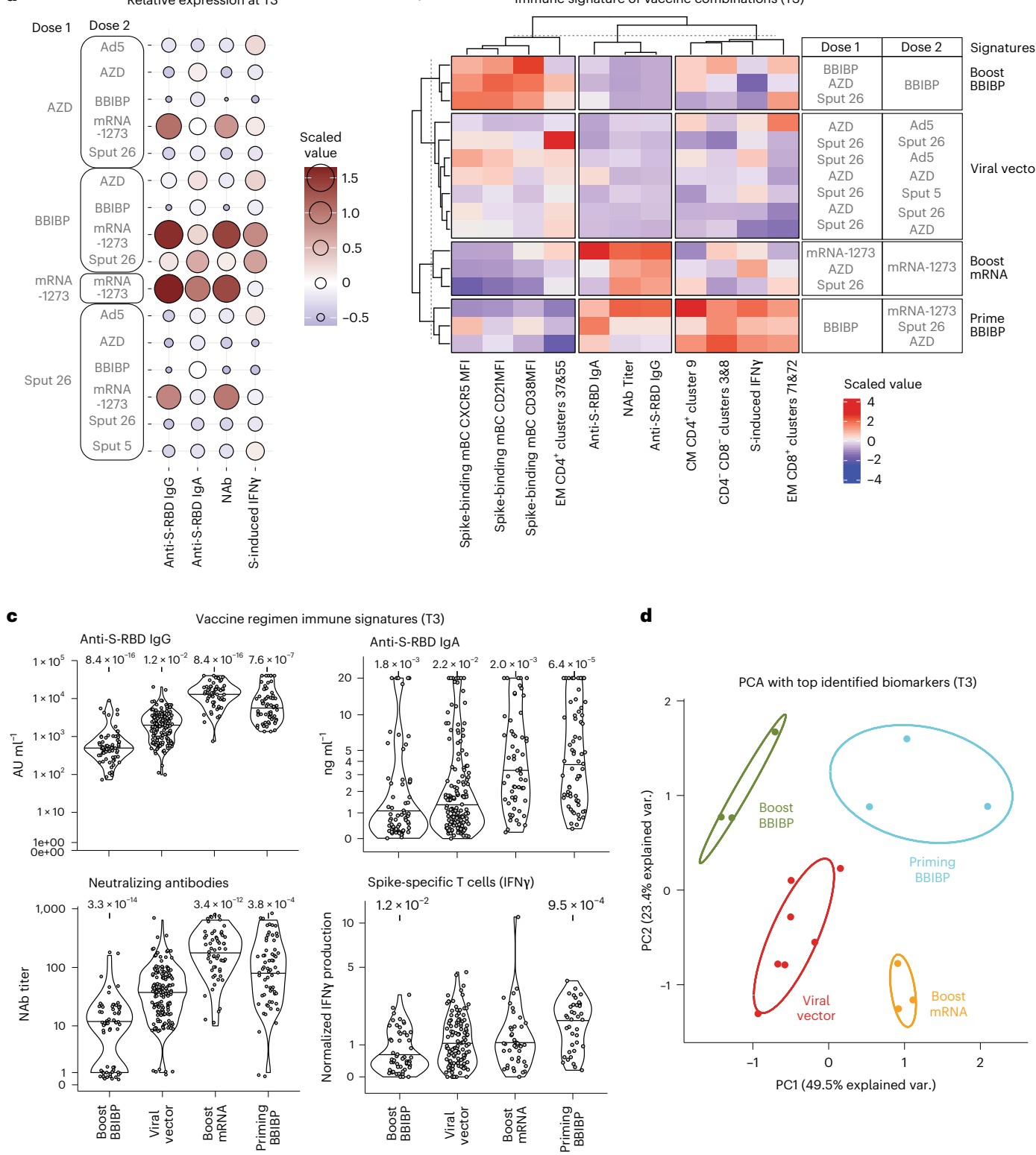

**Fig. 7 | Immune signatures of specific vaccine combinations. a**, Scaled and centered levels of neutralizing (NAb) and spike-specific (anti-S-RBD IgG) antibodies as well as SARS-CoV-2 spike peptide-induced T cell IFNγ production for each vaccine combination at T3 ($n = 347$). **b**, Scaled and centered values per column of the top humoral and cellular immune features for each immune signature, displayed in a heatmap with $k$-means clustering applied to the rows and columns ($n = 347$). **c**, Anti-S-RBD IgG levels, anti-S-RBD IgA levels, NAb titers and SARS-CoV-2 spike peptide-induced IFNγ production from all participants grouped by immune signature ($n = 347$). Large black dots depict the median of the group, and the vertical line spans the IQR. $P$ values indicate differences between the respective group and the overall mean of all participants and were calculated using the Mann–Whitney–Wilcoxon test and the Benjamini–Hochberg method to control for multiple hypothesis testing. Only significant $P$ values ($P < 0.05$) are displayed. **d**, Principal component analysis of the top humoral and cellular immune features for each immune signature at T3 ($n = 347$).

(Fig. 7b). Differences in immune signatures were striking between priming and boosting with BBIBP. The prime BBIBP signature was found to have a comparably high frequency of the CM CD4+ cluster 9, CD4−CD8− clusters 3&8 and the EM CD8+ clusters 71&72. On the contrary, the immune signature of boost BBIBP was characterized by spike-binding mBCs with a higher expression of CD21, which is associated with a resting mBC phenotype, while the immune signature of boost mRNA was characterized by lower CD21 expression by spike-binding mBCs, marking activated or atypical mBCs (depending on CD27 expression)[27–29]. In addition, we observed a lower expression of CXCR5 in the boost mRNA compared to the boost BBIBP group, a receptor for participating in germinal center reactions that is typically downregulated in non-classical mBC subsets[20,30–32]. We compared antibody production and cellular immune responses to SARS-CoV-2 among these four immune signature groups. The signature group 'boost mRNA' induced the highest anti-RBD-IgG and NAb response, while 'priming BBIBP' led to the strongest anti-RBD-IgA and antigen-specific T cell responses (Fig. 7b,c). Finally, to visualize their distinct characteristics, principal component analysis with the same 11 parameters shows the clear separation of the vaccine signatures (Fig. 7d). Our analysis thus allows the extraction of vaccine regimen-driven immune signatures, which are linked to immunogenicity. These results provide insights into the underlying immunological mechanisms of vaccine-induced immune responses.

## Discussion

The plethora of different COVID-19 vaccines available offers an unprecedented opportunity to study human immune responses to immunization. Here we present the most comprehensive head-to-head immunophenotyping comparison of vaccine protocols to date, covering adenoviral-vector, inactivated virus and mRNA platforms. We show that several heterologous vaccine combinations have similar or superior humoral and cellular immunogenicity compared to homologous regimens. In addition, we detected B and T cell phenotypes that correlated with the humoral and cellular immune response, respectively, and classified the 16 vaccine combinations into four distinct groups based on differing humoral and cellular immune signatures induced in vaccinated individuals.

Our finding that the heterologous regimens AZD/mRNA-1273 and AZD/Ad5 induced stronger humoral and cellular responses compared to AZD/AZD agrees with preliminary reports showing improved immunogenicity for AZD when combined with mRNA vaccines (mRNA-1273 or BNT162b2)[2,3,5]. In BBIBP-primed individuals, all heterologous combinations led to stronger immune responses than the homologous regimen. These findings corroborate the results of antibody analyses from the complete ECEHeVac cohort (which included three additional centers in Argentina)[18]. In general, the administration of mRNA-1273 as dose 2 clearly improved immunogenicity in all assessed conditions. Combinations such as AZD/Ad5, BBIBP/Sput-26 and BBIBP/AZD also proved to be highly immunogenic. Together, these results provide a strong rationale for the superiority of heterologous vaccine regimens against SARS-CoV-2 among non-mRNA-based vaccines, suggesting that this strategy could improve the efficiency of vaccination programs, particularly in regions with limited vaccine supply. Interestingly, the order of vaccine administration in heterologous regimens also appears to be important—in line with other studies[33,34]; we observed that BBIBP induced strong immune responses as dose 1 but not as dose 2.

SARS-CoV-2 spike-binding mBC phenotypes differed among the vaccine combinations—boosting with BBIBP induced a resting phenotype, while boosting with mRNA-1273 led to a switched-activated or atypical phenotype of SARS-CoV-2 spike-specific mBCs[24]. The exact mechanisms underlying these differences in specific mBC induction remain unclear, but differential interactions between antigen-specific B cells and membrane-bound versus soluble antigens may be responsible[35]. Furthermore, we found that boosting with BBIBP coincided with the highest proportions of CXCR5hiCXCR3lo

spike-specific mBCs, a phenotype reminiscent of naive B cells. As in our study, Zhang et al. reported a correlation between NAb titers and CXCR3 expression by SARS-CoV-2 spike-specific mBCs in individuals who received an adenovirus-based COVID-19 vaccine[36]. Atypical mBCs (CD19+CD21−CD27−) have been shown to be activated in response to membrane-associated antigen and to be able to present antigen to T cells as well as to differentiate into plasma cells[37]. A subset of CD21lo antigen-specific B cells has been described to transiently arise 14–28 d after influenza vaccination. These cells potentially emerge from germinal centers, are refractory to further germ center differentiation (as indicated by downregulation of CXCR and CXCR5 that are associated with trafficking to and within germinal centers) and primed to differentiate into long-lived plasma cells[20]. Our data indicate SARS-CoV-2 spike-specific mBCs with a comparable phenotype (CD21loCD38loCXCR5lo) to be associated with stronger antibody response, suggesting that potent vaccine regimens induce this subset that then gives rise to antibody-producing plasma cells. However, BBIBP-boosted individuals showed a higher expression of CD21, CXCR4 and CXCR5 as well as a predominantly IgM+ spike-binding memory B-cell phenotype, resulting in a weaker antibody response but hypothetically retaining the ability of the antigen-specific memory B cells to re-enter germinal centers and thus a potentially broader immune response.

mBCs are of particular interest in the context of new SARS-CoV-2 variants, as these cells undergo fewer somatic hypermutations than plasma cells and are potentially more flexible in responding to different viral subtypes[38]. IgM+ mBCs tend to migrate to B cell follicles and re-initiate germinal center reactions upon rechallenge, thus potentially increasing the breadth of the antibody response; meanwhile, IgG+ mBCs preferentially differentiate into plasma cells to rapidly induce specific antibody production[21]. In our study, IgM expression of spike-binding mBCs was higher in BBIBP-primed individuals at T3, independent of the booster vaccine. Conversely, priming with the other vaccines resulted in the generation of primarily IgG+ spike-binding mBCs. IgM+ mBCs have been shown to provide a broader recognition of viral variants, as seen in a mouse model of immunization with dengue virus variant proteins[35,39]. In humans, a study of homologous versus heterologous vaccine combinations showed that BBIBP/mRNA-1273 had the highest neutralizing activity against the Omicron variant[18]. Thus, a vaccine response that induces a higher IgM expression among mBCs appears to deliver a broader immune response, potentially increasing immunity against viral variants. In addition, we found mRNA-1273/mRNA-1273 to most strongly induce an IgA-polarized immune response. IgA antibodies were demonstrated to dominate the early humoral response to SARS-CoV-2 infection and to have stronger neutralization capacities compared with IgG[40] and were found even in the absence of IgG in individuals after asymptomatic SARS-CoV-2 infection[41]. We speculate that differences in the cytokine response to the vaccine regimens may result in differences in IgA/IgG isotype polarization of the B cell response. For example, TGFβ is known to be essential for T-cell-dependent class-switching to IgA[42].

Regarding the induction of spike-specific T cells, priming with BBIBP proved to be an optimal base for a strong IFNγ response. We found two T cell clusters (CD4−CD8− clusters 3&8) that consistently expanded to a greater extent in the BBIBP-primed groups. These clusters were similar to the canonical CD4−CD8− T cell subsets but displayed higher expression of CD27 and CD45RA, markers associated with naive T cells, and lower expression of killer-like-lectin receptor 1 (KLRG1), which has been described as a marker of final differentiation and immunosenescence[43]. While KLRG1hi T cells are generally considered short lived, it has been shown that KLRG1 can be downregulated (ex-KLRG1), giving rise to multiple memory populations that contribute to an effective antiviral response[44]. CD4−CD8− (double-negative) T cells make up approximately 3–5% of circulating T cells[45]. They can originate from double-positive thymocytes by downregulation of CD4 and CD8 in the peripheral blood[46]. Such double-negative T cells are restricted to

either MHC I or MHC II molecules and have been shown to share functions with different subtypes of effector T cells, such as regulatory T cells, T helper (Th) cells or cytotoxic T cells. They were proposed to contribute to both innate and adaptive immune responses by modulating the functions of macrophages, CD8[+] T cells and B cells. As for viral infections, Th-like double-negative T cells in natural nonhuman hosts of simian immunodeficiency virus were shown to produce Th1, Th2 and Th17 cytokines and they have been associated with suppression of disease progression[45]. Of note, our data do not show whether the described T cell clusters are themselves responsible for the observed IFNγ production after stimulation with spike peptides, as the clusters were identified through correlative analyses. However, they could nonetheless serve as biomarkers for the cellular response after COVID-19 vaccination. In summary, our results show that priming with the inactivated virus vaccine leads to specific characteristics in the B and T cell compartments that are maintained even after a booster dose of a different vaccine. Differences between vaccines that might contribute to the heterogenicity of antibody, B and T cell responses might concern the version of the SARS-CoV-2 spike protein used, adjuvants (such as aluminum hydroxide in BBIBP[47]) or the presentation of the antigen (as part of an inactivated virus in BBIBP or produced by host cells in mRNA/viral-vector based vaccination). Also, the inactivated virus vaccine offers a fixed amount of the spike protein, whereas mRNA and adenoviral-vector-based vaccines may produce variable quantities of the antigen. The exact mechanisms however remain to be explained.

While clinical studies are needed to determine whether the comparatively higher immunogenicity of specific vaccination regimens translates to better protection from SARS-CoV-2, the immune signatures described in our study can serve as a guide for vaccine development. Also, our results suggest that heterologous boosting might improve protection in individuals who have received one of the less immunogenic COVID-19 vaccines. The data suggest that heterologous vaccination may be in general superior and therefore should be considered as a valuable strategy for future vaccines against pathogens and cancer.

A limitation of this study is the difference in time intervals between doses 1 and 2 for some groups. A 12-week interval was previously shown to induce higher NAb titers, but weaker T cell responses, compared to a 4-week interval for homologous and heterologous combinations of AZD and BNT162b2 (ref. 48). Nonetheless, some of the regimens with the shortest dosing intervals in our cohort (for example, mRNA-1273/mRNA-1273) displayed the highest NAb titers among all the groups, and the intervals were comparable among individuals receiving the same dose 1. Additionally, we did not evaluate the real-world efficacy of the different vaccine combinations and the persistence of their immune responses over time. Lastly, while high age has been shown to have an influence on antibody titers[49], which may have a minor influence in this study as well, the correlation with the immunophenotype at the single-cell level was performed with PBMC samples spanning a patient group of 18–59-year-old individuals.

Our results indicate that different types of vaccines induce distinct cellular immune responses. With this knowledge, combinatorial approaches might be able to exploit the strengths of different vaccine techniques. The identified characteristics in the immune response to different types of vaccines expand our understanding of vaccine-induced immunity and may be highly valuable for the future development and testing of vaccines against infectious diseases as well as cancer.

## Online content

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

[1]Institute of Experimental Immunology, University of Zurich, Zurich, Switzerland. [2]Facultad de Ciencias Médicas, Instituto de Virología 'Dr. J. M. Vanella' Universidad Nacional de Córdoba, Córdoba, Argentina. [3]Facultad de Ciencias Químicas, Departamento de Bioquímica Clínica, Universidad Nacional de Córdoba, Córdoba, Argentina. [4]Consejo Nacional de Investigaciones Científicas y Técnicas (CONICET), Centro de Investigaciones en Bioquímica Clínica e Inmunología (CIBICI), Córdoba, Argentina. [5]Ministerio de Salud de la Nación, Buenos Aires, Argentina. [6]Secretaría de Prevención y Promoción de la Salud, Ministerio de Salud de la Provincia de Córdoba, Córdoba, Argentina. [7]Consejo Nacional de Investigaciones Científicas y Técnicas (CONICET), Buenos Aires, Argentina. [8]Present address: Facultad de Ciencias Químicas, Departamento de Bioquímica Clínica, Universidad Nacional de Córdoba, Córdoba, Argentina. [9]Present address: Consejo Nacional de Investigaciones Científicas y Técnicas (CONICET), Centro de Investigaciones en Bioquímica

Clínica e Inmunología (CIBICI), Córdoba, Argentina. [10]These authors contributed equally: Nicolás Gonzalo Nuñez, Jonas Schmid, Laura Power. [11]These authors jointly supervised this work: Sandra Gallego, Gabriel Morón, Laura Cervi, Eva V Acosta Rodriguez, Belkys A Maletto, Mariana Maccioni, Burkhard Becher. *Lists of authors and their affiliations appear at the end of the paper. ✉e-mail: nicolas.nunez@unc.edu.ar; mariana.maccioni@unc.edu.ar; becher@immunology.uzh.ch

## InmunoCovidCba

**Fabio Cerbán[4], Laura Chiapello[4], Carolina Montes[4], Cristina Motrán[4], Jeremías Dutto[4], Laura Almada[4] & Lucía Boffelli[4]**

## InViV working group

**Lorena Spinsanti[4], Adrián Díaz[4], María Elisa Rivarola[4], Javier Aguilar Bioq[4] & Mauricio Beranek[4]**

## Methods

### Sample donors

Volunteers (age range: 18–82 years) were enrolled in a randomized, open Phase IIB clinical trial (ECEHeVac, NCT04988048) aimed at comparing the immunogenicity and reactogenicity of heterologous and homologous vaccination regimens available in Córdoba, Argentina. The study received ethical approval from the Registro Provincial de Investigación en Salud (Provincial Registry of Health Research, REPIS-Cba 4371). The study was conducted in accordance with the guidelines of Good Clinical Practice (ICH 1996) and the principles of the Declaration of Helsinki. No compensation was provided to the study participants.

### Eligibility criteria

Eligible participants were healthy volunteers older than 18 years who had received a first dose of the AZD, BBIBP, Sput-26 or mRNA-1273 vaccine 30–120 d before the enrollment date. Exclusion criteria were immunocompromised status with underlying disease or immunosuppressive treatment; pregnancy and lactation; having received a major surgical intervention in the 30 d before the enrollment date; having had a severe allergic reaction (anaphylaxis) to any vaccine; having a visceral disease that lead to disability (heart failure, kidney failure, respiratory failure, liver failure, intestinal malformations, electro-dependence or having had a visceral transplant less than 2 years previously) and having had COVID-19 (symptomatic or asymptomatic). Additionally, all participants were tested for anti-SARS-CoV-2 nucleocapsid IgG via ELISA on T1 and in case of positive results excluded from the study (except for those participants that had been vaccinated with BBIBP as the first dose).

### Randomization, consent and follow-up

Participants were randomized with equal group allocation to determine the vaccine used as dose 2, and the participant and healthcare personnel in charge of vaccination were informed of the result. Each participant provided written informed consent to be included in the study. All participants filled out a questionnaire to verify personal data and health history. In this paper, the day of dose 2 is referred to as timepoint 1 (T1). The participants were observed for 15–20 min after inoculation. After T1, telephone and face-to-face monitoring of each participant was carried out for up to 6 months. The data obtained were recorded in the national health information system (www.https://sisa.msal.gov.ar/sisa/#sisa).

### Serum, plasma and PBMC collection

Whole blood samples were collected from participants at the day of dose 2 administration immediately before vaccination (T1) and at 14 ± 2 (T2) and 28 ± 1 (T3) days after dose 2. Sera and plasma were obtained from the whole blood at each timepoint. PBMCs (799 samples acquired; after exclusion of samples not meeting the eligibility criteria 754 samples from 407 individuals were analyzed; 209 female and 198 male; age range: 18–59 years) were isolated from samples at T1 and T3 via density-gradient sedimentation using Ficoll-Paque PLUS (GE Healthcare). Isolated PBMCs were cryopreserved in heat-inactivated FBS (Natocor) containing 10% DMSO (Sigma-Aldrich) and stored in liquid nitrogen until use. Plasma samples (1,491 samples from 497 individuals; 252 female and 245 male; age range 18–82 years) were used for SARS-CoV-2 neutralization assays, sera were used for IgG anti-SARS-CoV-2 assays, and PBMC samples were used for flow cytometry and functional T cell assays.

### Detection of anti-SARS-CoV-2 IgG and IgA antibodies

IgG antibodies against the SARS-CoV-2 nucleocapsid protein were qualitatively detected on T1 using a chemiluminescent microparticle immunoassay (CMIA; ARCHITECT SARS-CoV-2 IgG; 6R86, Abbott) that relies on an assay-specific calibrator to report a ratio of specimen to calibrator absorbance (S/C). The interpretation of the result is determined by an index value, which is a ratio over the threshold value. An index (S/C) of <1.4 was considered negative and ≥1.4 was considered positive. IgG antibodies against the RBD of the S1 subunit of the SARS-CoV-2 spike protein were evaluated in sera samples collected on T1, T2 and T3 by a quantitative CMIA assay (AdviseDx SARS-CoV-2 IgG II; 6S60, Abbott). Sera with values ≥50.0 Arbitrary units per milliliter were considered positive. IgA antibodies against SARS-CoV-2 S1 RBD were measured in sera collected at T3 (samples from 349 individuals; 178 female and 171 male; age range: 18–59 years) using ELISA (SARS-CoV-2 S1 RBD IgA ELISA kit; 4A257R, ImmunoDiagnostics). Sera were diluted tenfold and the measurements were performed according to the manufacturer's instructions. The optical density (OD) of a blank well was subtracted from the OD of the samples measured at 450 nm. Measured sample values above 20 ng ml$^{-1}$ were set to >20 ng ml$^{-1}$. The cut-off between IgA positive and negative responders (cut-off: 0.32 ng ml$^{-1}$) was determined using a SARS-CoV-2 unexposed prepandemic serum sample.

### Detection of neutralizing antibodies against live SARS-CoV-2

Plasma samples from T1, T2 and T3 were tested for their ability to neutralize wild-type SARS-CoV-2 B.1 (hCoV-19/Argentina/PAIS-G0001/2020, GISAID accession ID: EPI_ISL_499083) using the plaque reduction neutralization test as previously described[50]. Briefly, this test was performed with Vero 76 cells (ATCC CRL-1587) that were seeded in 24-well plates 48 h before infection. Plasma samples were heat-inactivated by incubation at 56 °C for 20 min and centrifuged at 3,000$g$ 30 min before use. Treated samples were diluted twofold, and an equal volume of virus stock containing 100 plaque-forming units (PFU) was added to each corresponding well until reaching final dilutions ranging from 1:10 to 1:320. Cells were incubated with 0.5% agarose with DMEM supplemented with 2% FBS for 4 d at 37 °C in a 5% $CO_2$ incubator. After 4 d, cells were fixed and inactivated using a 10% formaldehyde/PBS solution and stained with 1% crystal violet. NAb titers corresponded to the maximum dilution of plasma that neutralized 80% of the PFU, compared with PFU from the viral controls included in the test.

### IFNγ ELISpot assay

IFNγ ELISpot analysis was performed using microplates precoated with monoclonal IFNγ-specific antibodies (Human Interferon Gamma ELISPOT Kit; Abcam). PBMCs (250,000 cells per well) were separately stimulated for 18 h with peptide pools (15mers with 11 amino acids overlap, PepMix SARS-CoV-2 spike Glycoprotein and NCAP, JPT Peptide Technologies) at a concentration of 1 μg ml$^{-1}$. Tests were conducted with a negative (DMSO, Sigma-Aldrich) and positive control (Dynabeads Human T-Activator CD3/CD28, Gibco) for each sample. According to the manufacturer's protocol, we added a biotinylated anti-IFNγ detection antibody, followed by a streptavidin-AP conjugate and a 5-bromo-4-chloro-3-indolyl phosphate/nitro blue tetrazolium substrate (all from the Human Interferon Gamma ELISPOT Kit) to visualize bound IFNγ.

Plates were scanned using an AID Classic ELISpot Reader, and spots were counted with the AID ELISpot software version 7.0 (AID Autoimmun Diagnostika GmbH), following guidelines for the automated ELISpot evaluation[51]. Samples were excluded if the negative control wells had more than 39 or the positive control wells fewer than 40 spots. Spot counts were multiplied by 4 to evaluate spots per million cells and normalized by dividing by the well saturation of the positive control (spots per million cells/positive control well saturation) for each sample. We used repetitive control samples for both acquisition rounds to control for batch effects. The positive cut-off threshold was calculated by taking the mean of the normalized IFNγ responses of the negative control (DMSO) across all groups.

### Flow cytometry and data acquisition

Biotinylated full-length spike and nucleocapsid proteins (R&D Systems) were multimerized with streptavidin (SA)-BV421 (200 ng spike with 20 ng SA; ~4:1 molar ratio) and SA-PE-Cy5 (50 ng nucleocapsid with 14 ng SA;

~4:1 molar ratio), respectively, for 1 h[23,52]. Staining with SA-BV421 and SA-PE-Cy5 alone (without biotinylated spike or nucleocapsid protein) as well as fully stained SARS-CoV-2 unexposed prepandemic samples were used as controls. For the spike-binding mBC and T cell panels, 1.5 × 10[6] and 1.0 × 10[6] PBMCs, respectively, were washed with PBS and blocked using Human TruStain FcX and True-Stain Monocyte Blocker (BioLegend). First, for the spike-binding mBC panel, cells were incubated for 30 min at 37 °C with the tetramers and antibodies listed in Supplementary Table 2. For both the spike-binding mBC and T cell panels, cells were stained for 25 min at 4 °C with the antibodies listed in Supplementary Table 2 or Supplementary Table 3, respectively. Following surface staining, cells were fixed with 2% PFA or with Foxp3/transcription factor fixation/permeabilization solution (eBioscience) for 15 or 40 min at 4 °C, respectively, for the spike-binding mBC and T cell panels. For the T cell panel, cells were then stained overnight at 4 °C with the antibodies listed in Supplementary Table 3, all diluted in 1× permeabilization buffer (eBioscience). Data were acquired with Cytek Aurora flow cytometers and preprocessed using FlowJo software version 10 (BD Bioscience).

### High-dimensional flow cytometry data analysis
For high-dimensional flow cytometry analysis, dead cells, doublets and cells stained by fluorochrome aggregates were excluded from the analysis via manual gating using FlowJo software. The gating strategy is shown in Extended Data Fig. 7a,b. Datasets of different batches were corrected using the CytoNorm R package[53]. To obtain an unbiased overview, we systematically reduced cytometry data to two dimensions by applying UMAP[54] (umap R package[55]) to stochastically selected cells. All cells were clustered using the FlowSOM algorithm (FlowSOM R package[56]) in conjunction with consensus clustering (ConsensusClusterPlus R package[57]) and were subsequently manually annotated into different clusters with distinct phenotypes in terms of median fluorescence intensity of the selected surface marker. The expression of each marker across all samples was min–max normalized to the range 0–1. The main R script was run as described in ref. 58.

### Statistical analysis
Comparisons of continuous variable means between groups or to the overall mean of all participants were performed using the Wilcoxon–Mann–Whitney test and corrected for multiple hypothesis testing with the Benjamini–Hochberg method (ggpubr R package[59]); these tests were two-tailed and performed on unpaired data. To calculate log-transformed data, we added 1 to the value before taking the $\log_2$. Where the $y$-axis was scaled, the pseudo_log_trans(base = 2) function was used (scales R package[60]). Spearman's rank correlation coefficients ($r_s$) between continuous variables were calculated with the Hmisc R package[61], which approximates $P$ values by using asymptotic $t$ distributions. Scaled expression plots were scaled (values were divided by the standard deviation of the group) and centered (the group mean was subtracted from the values) per marker or feature. Differential expression was calculated by subtracting the expression level of the canonical T cell subset from the expression level of the T cell subcluster of interest. Fold change was calculated by dividing the participant value at T3 by the mean group value at T1. $K$-means clustering was performed on the rows and columns of heatmaps (ComplexHeatmap R package[62]). $P < 0.05$ was considered statistically significant. All statistical analyses were performed using R version 4.0.1 (R Core Team 2020).

### Reporting summary
Further information on research design is available in the Nature Portfolio Reporting Summary linked to this article.

### Data availability
This study did not generate new reagents. Data are available in a public repository (https://doi.org/10.5281/zenodo.7734088). Source data are provided with this paper.

### Code availability
This study did not generate new codes. The codes that support these findings have been previously described[53–62].

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

### Acknowledgements
We thank the study participants who contributed to this work. We thank C. D. Anderfuhren for technical support. We thank the authorities of Facultad de Ciencias Químcas-UNC and J. Zarzur (Fundación para el Progreso de la Medicina) for their support to the project. We thank N. Ponce, P. A. Icely, G. Furlan and N. Maldonado for their technical assistance. This project has received funding from the European Research Council (ERC) under the European Union's Horizon 2020 research and innovation program grant agreement No 882424, the Swiss National Science Foundation (31CA30_195883 and CRSII5_183478 to B.B.), The Loop Zurich—Medical Research Center, the Vontobel foundation, Agencia Nacional de Promoción Científica y Técnica (PICT 2021 CAI-I 00051) and Fondo para la Investigación Científica y Tecnológica (FONCyT) (PICT IP COVID 19-464, 2020); School of Medical Sciences, National University of Córdoba, Argentina and the Ministry of Health of Córdoba Province, Argentina. N.G.N. is a recipient of a University Research Priority Program (URPP) postdoctoral fellowship and received funding from the RAÍCES program (MINCyT). S. Kreutmair (442457282) and T.W. (WE 6945/1-1) are recipients of a postdoctoral research fellowship of the German Research Foundation (DFG). The schematic representation of the study protocol was created with a full license of BioRender. We thank D. Ackerman from Insight Editing London for critical review and editing of the manuscript.

## Author contributions

M.E.P., J.M.C. and C.V. designed the clinical trial (ECEHeVac, NCT04988048). D.C. and G.B. led the clinical trial in Córdoba (REPIS-Cba 4371) and provided funding for the generation of the PBMC biobank. L.L. and P.C. were in charge of the randomization, participant data collection and follow up of donors. C.M., L.O., A.G., C.S. and the ImmunoCovidCba group collected and processed PBMC samples. E.V.A.R., L.C., B.M., G.M. and M.M. supervised the generation of the PBMC biobank. G.C., E.R., S.B., B.K. and the InViV working group performed antibody and neutralizing titers measurements. N.G.N., J.S. and C.A. conceptualized and designed flow cytometry and ELISpot experiments. N.G.N., J.S., C.A., S. Kreutmair, S. Krishnarajah, S.U., J.C.K., S.C.P., F.I., M.S., J.V.V., C.H.M., A.S., T.W., M.L. and D.V. performed experiments. L.P., N.G.N. and J.S. analyzed the data. J.S., N.G.N., L.P. and S. Krishnarajah wrote the original draft. J.S., N.G.N., L.P., S. Krishnarajah, B.B., C.M., G.M., L.C., E.V.A.R., B.A.M., M.M., S. Kreutmair, S.U., F.I. and T.W. reviewed and edited the manuscript. B.B., S.G., G.M., L.C., E.V.A.R., B.A.M. and M.M. supervised and funded the study. C.A., S. Krishnarajah, S. Kreutmair and S.U. contributed equally. All authors read and approved the manuscript.

## Funding

## Competing interests

All authors declare no competing interests.

## Additional information

**Extended data** is available for this paper at https://doi.org/10.1038/s41590-023-01499-w.

**Correspondence and requests for materials** should be addressed to Nicolás Gonzalo Nuñez, Mariana Maccioni or Burkhard Becher.

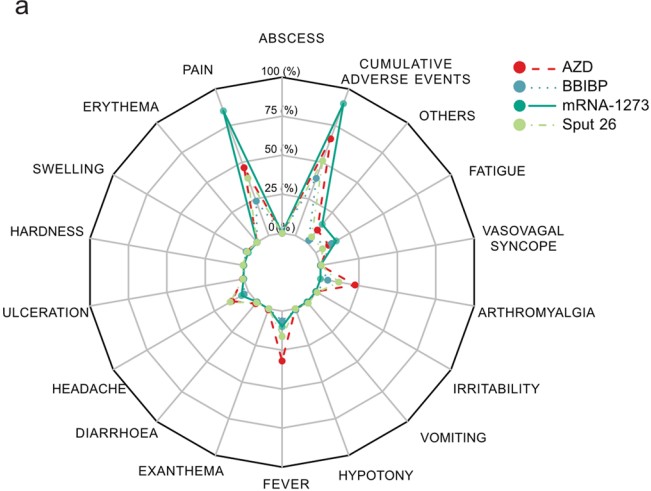

**Extended Data Fig. 1 | Local and systemic adverse events following vaccination.** Reported adverse events and percentage of participants experiencing them are reported per vaccination group **a**, after the first vaccine dose and **b**, after the second dose (*n* = 497). Shaded areas indicate the homologous vaccine combinations.

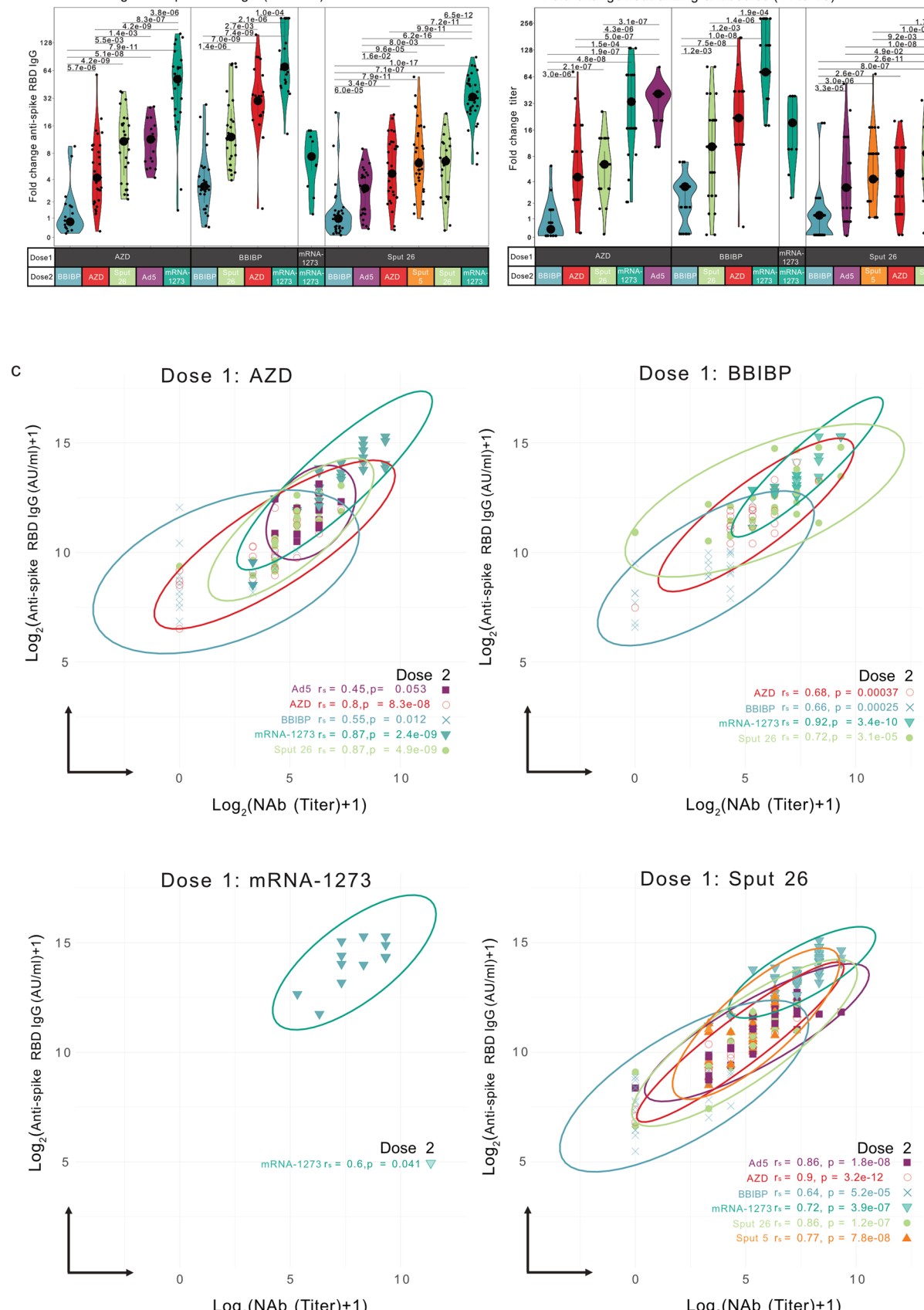

**Extended Data Fig. 2 | See next page for caption.**

**Extended Data Fig. 2 | Anti-S-RBD IgG levels and neutralizing antibody titers are correlated and respond differently to various vaccine combinations.** Fold change in **a**, anti-S-RBD IgG levels ($n = 420$) and **b**, neutralizing antibody titers ($n = 421$) at T3 compared to the mean per group at T1. Large black dots depict the median of the group, and the vertical line spans the interquartile range. P-values were calculated using the Mann-Whitney-Wilcoxon test and the

Benjamini-Hochberg method to correct for multiple hypothesis testing. Only significant $P$ values ($P < 0.05$) are displayed. **c**, Correlation between the anti-S-RBD IgG levels and neutralizing antibody titers for every group ($n = 420$). The log transformation was performed by adding 1 to the value before taking the $\log_2$. The Spearman's rank correlation coefficients ($r_s$) and $P$ values ($P$) are indicated.

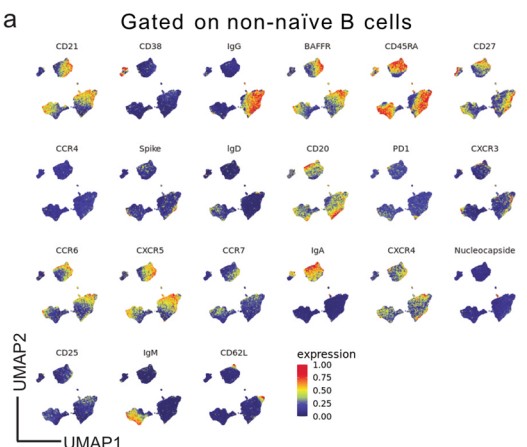

**a** Gated on non-naïve B cells

**b** Fold change in frequency of spike-binding mBCs (T1 to T3)

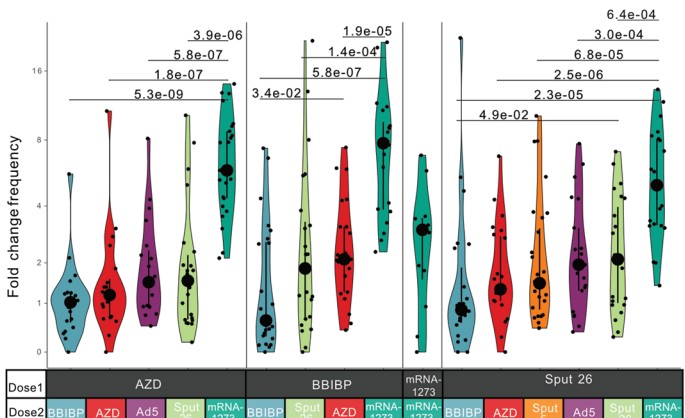

**c** Correlation of B cell subset frequencies with antibody response (T3)

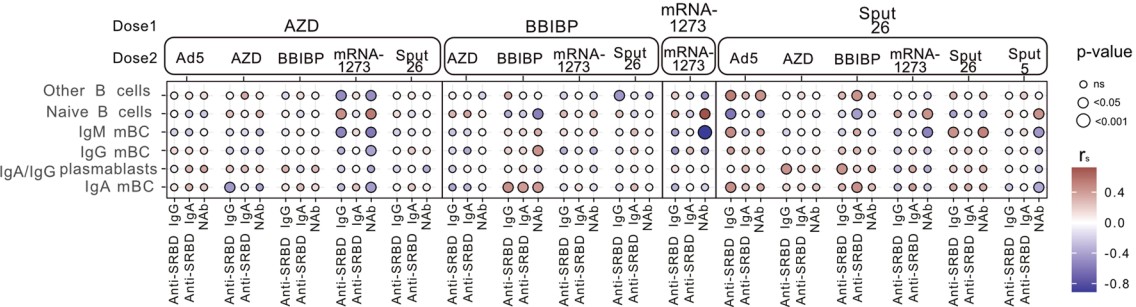

**d** Correlation of spike-binding mBC marker expression level with antibody response (T3)

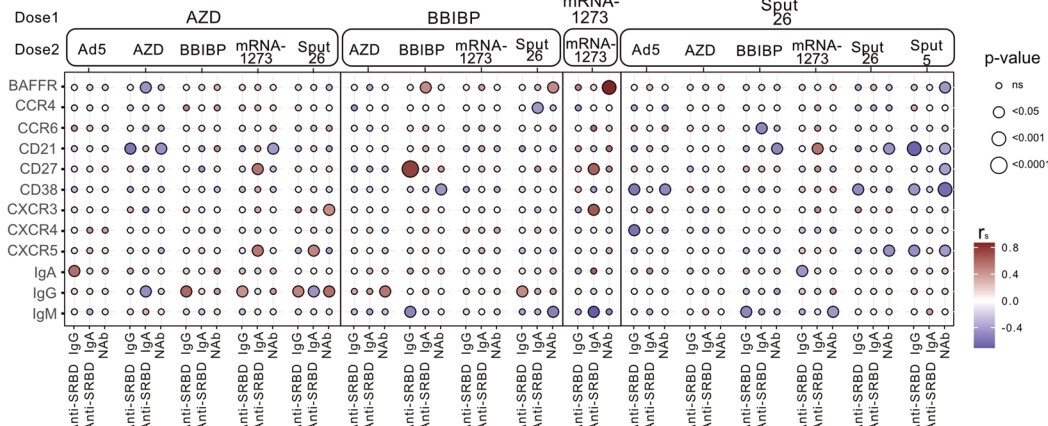

**e** Neutralizing antibody status **f** Anti-S-RBD IgA status

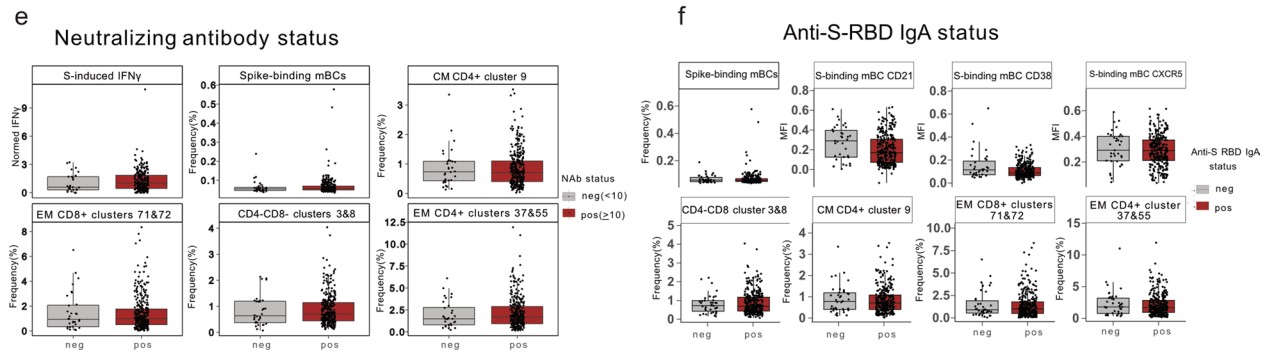

**Extended Data Fig. 3 | See next page for caption.**

**Extended Data Fig. 3 | Analysis of the B cell compartment after vaccination.**
**a**, UMAP showing the distribution of markers expressed by non-naïve B cells
(IgD⁻/IgM⁻) among all vaccine groups ($n = 799$). **b**, Fold change of spike-binding
mBC frequencies at T3 compared to the mean per group at T1 ($n = 347$). Large
black dots show the median, and the vertical line spans the interquartile range.
P-values were calculated using the Mann-Whitney-Wilcoxon test between
groups with the same dose 1 and the Benjamini-Hochberg method to correct for
multiple hypothesis testing. Only significant $P$ values ($P < 0.05$) are displayed.
**c**, Correlation among B cell subsets and antibody responses (anti-S-RBD IgG
levels, anti-S-RBD IgA levels and neutralizing antibody titers) ($n = 347$).
**d**, Correlation among spike-binding mBC marker expression levels and antibody

responses. Color indicates the Spearman's rank correlation coefficient ($r_s$),
and the circle size indicates the $P$ value ($n = 347$). **e**, Participants were classified
as negative ($<10$) or positive ($\geq10$) for neutralizing antibodies ($n = 347$).
**f**, Participants were positive ($>0.32$ ng/ml) or negative for anti-S RBD IgA
($n = 345$). (**e**,**f**) The means of each listed parameter were compared between
negative and positive responders using the Mann-Whitney-Wilcoxon test
and corrected for multiple hypothesis testing with the Benjamini-Hochberg
method. Boxes bound the interquartile range (IQR) divided by the median, and
Tukey-style whiskers extend to a maximum of $1.5 \times$ IQR beyond the box. Dots are
participant data points.

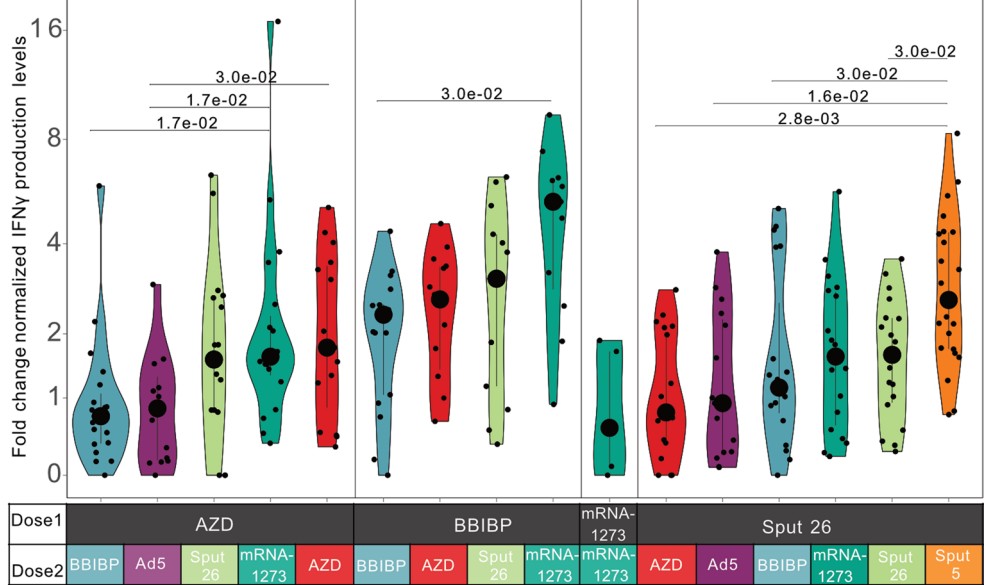

**Spike-specific T cells (IFNγ) fold change (T3)**

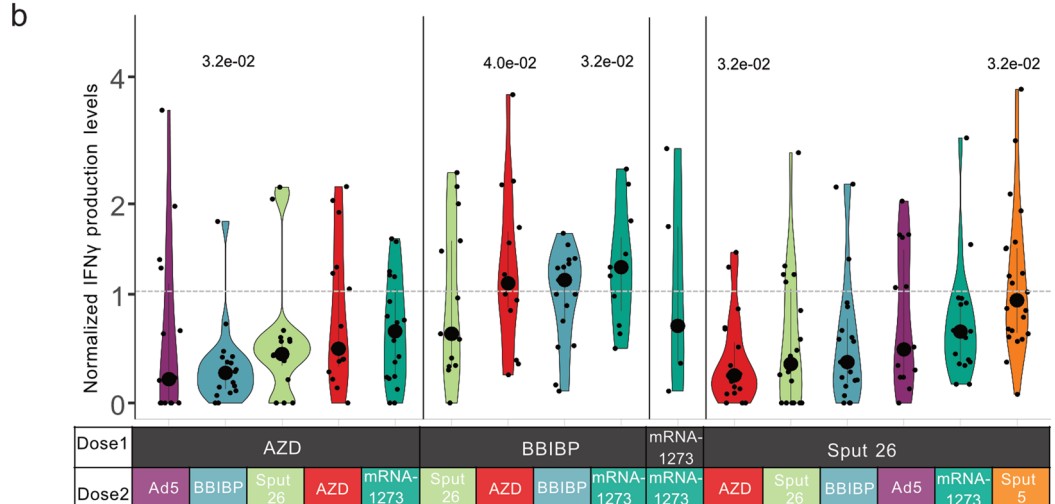

**Nucleocapsid-specific T cells (IFNγ) at T3**

**Extended Data Fig. 4 | Spike- and nucleocapsid-specific cellular IFNγ responses to antigen re-encounter after different prime-boost vaccine combinations. a**, Fold change (value at T3/group mean at T1) of the IFNγ response after stimulation of PBMCs with SARS-CoV-2 spike peptide pool, measured by ELISpot assay. Statistical tests were performed between groups with the same dose 1 (*n* = 255). **b**, Normalized IFNγ responses at T3 after stimulation of PBMCs with SARS-CoV-2 nucleocapsid peptide pool, measured by ELISpot assay (*n* = 254). The horizontal line indicates the cut-off threshold. *P* values indicate differences between the respective group and the overall mean of all participants. (a-b) P-values were calculated using the Mann-Whitney-Wilcoxon test and the Benjamini-Hochberg method to correct for multiple hypothesis testing. Only significant *P* values (*P* < 0.05) are displayed. Large black dots depict the median of the group, and the vertical line spans the interquartile range.

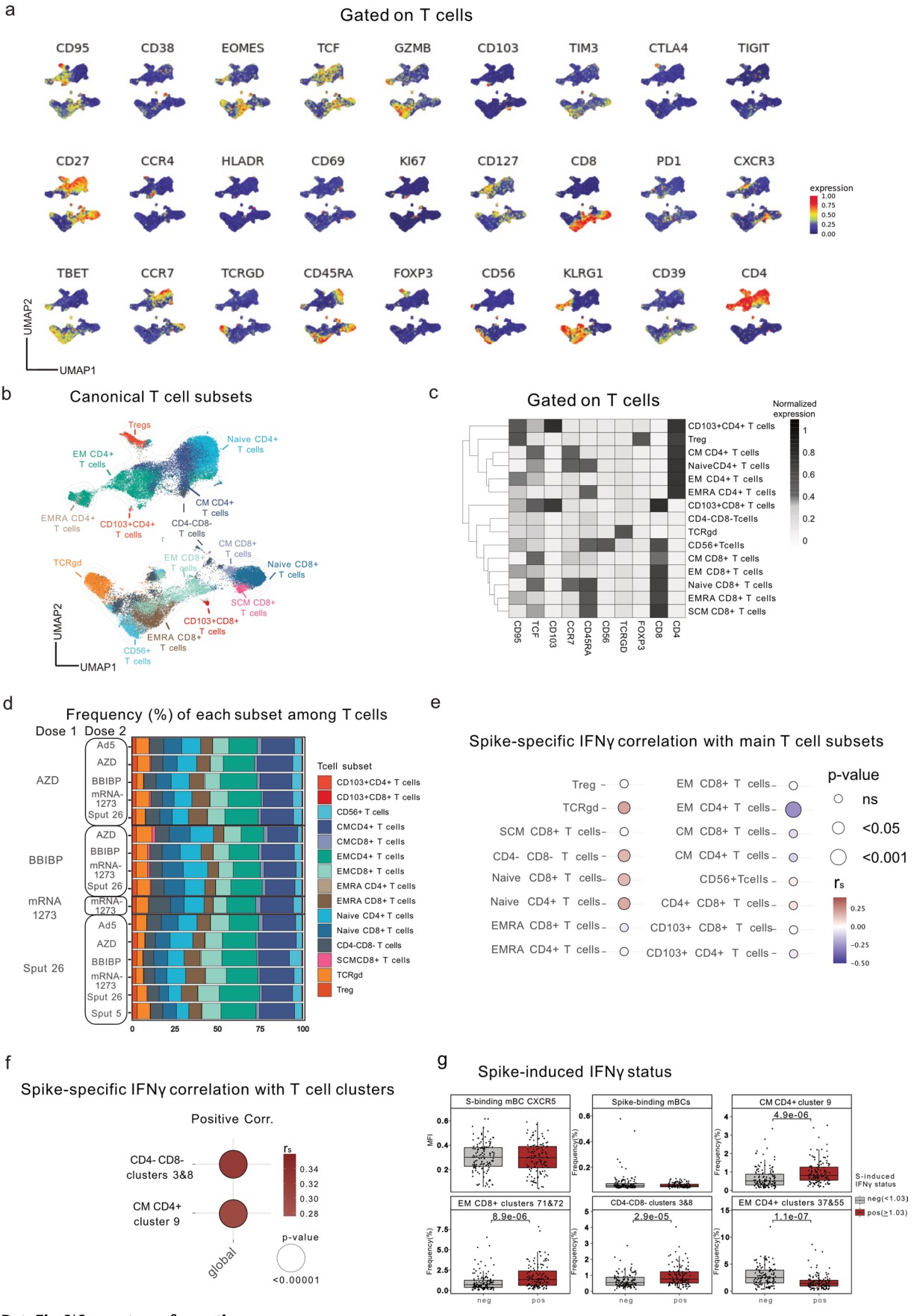

**Extended Data Fig. 5 | See next page for caption.**

**Extended Data Fig. 5 | Canonical T cell subsets and T cell subclusters related to the spike-specific IFNγ response. a**, UMAP showing the distribution of markers expressed by T cells among all vaccine groups ($n = 799$). **b**, UMAP showing the FlowSOM-guided manual metaclustering of T cells (CD3 + ) for all vaccine groups combined. **c**, Heatmap showing the median intensity of normalized marker expression (range 0-1) for each canonical T cell subset ($n = 799$). **d**, Frequencies of canonical T cell subsets relative to the total T cells for each vaccine regimen ($n = 347$). **e**, Spearman's rank correlations ($r_s$) of spike peptide-induced T cell response (IFNγ release) with the frequencies of (e) the canonical T cell subsets and **f**, T cell subclusters ($n = 255$). **g**, Participants were classified as negative (<1.03, the cut-off threshold) or positive (>1.03) for IFNγ response ($n = 255$). The means of each listed parameter were compared between negative and positive responders using the Mann-Whitney-Wilcoxon test and corrected for multiple hypothesis testing with the Benjamini-Hochberg method. Only significant $P$ values ($P < 0.05$) are displayed.

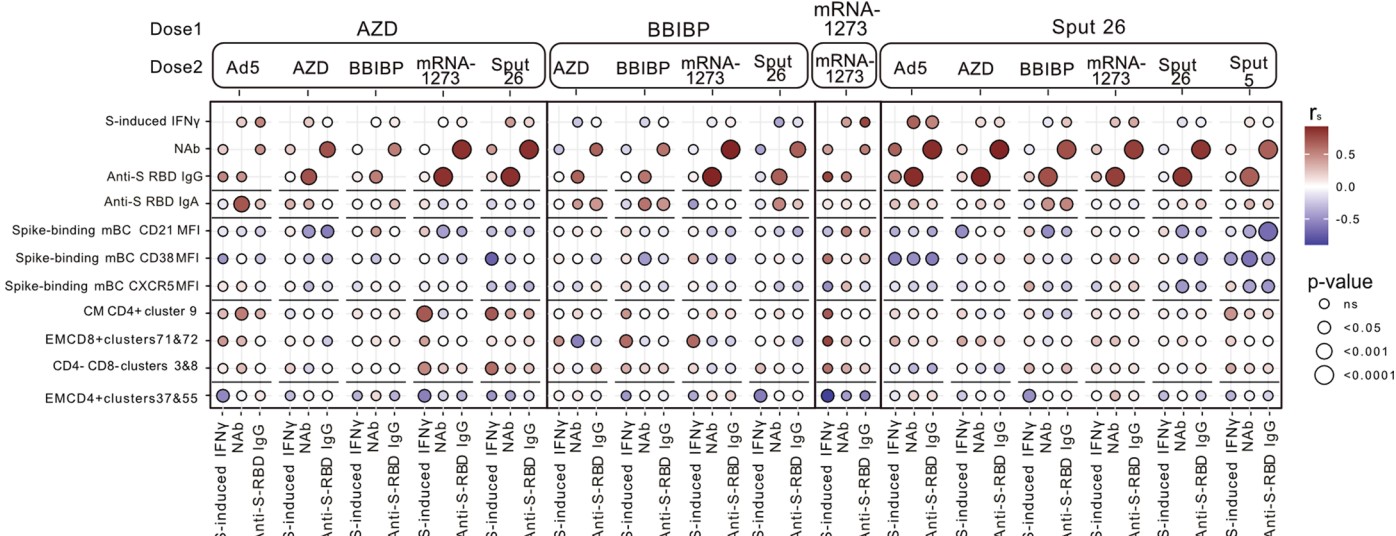

**Extended Data Fig. 6 | Immune parameters associated with positive and negative humoral and cellular responses.** Spearman's rank correlations ($r_s$) of the top humoral and cellular immune features with the neutralizing antibody (NAb) titers, anti-S-RBD IgG levels and SARS-CoV-2 spike peptide- induced T cell IFNγ response ($n = 347$).

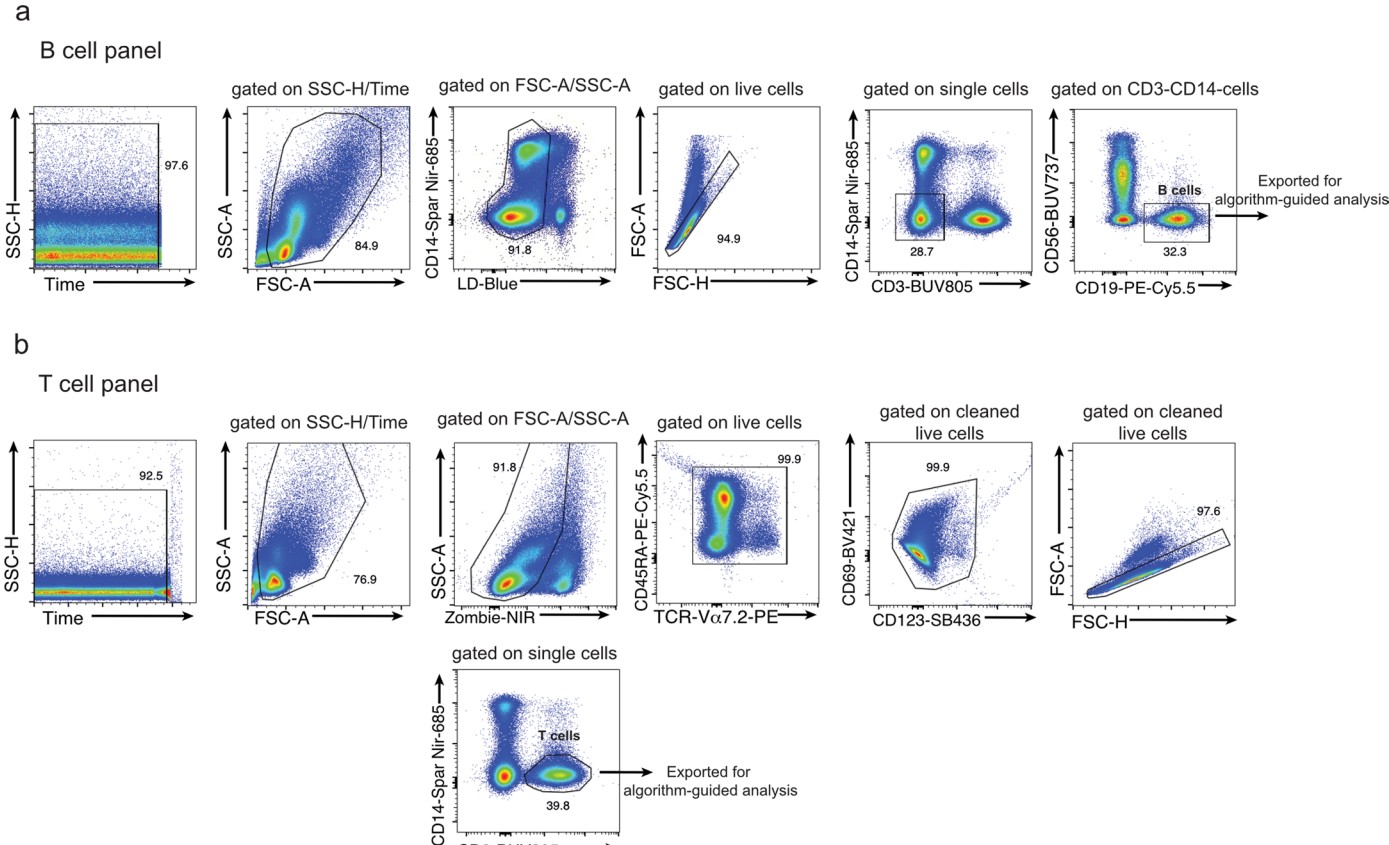

**Extended Data Fig. 7 | Data cleaning for high dimensional flow cytometry data analysis.** Representative flow cytometry gating strategy for data cleaning of **a**, B and **b**, T cell panels.

# Reporting Summary

## Statistics

For all statistical analyses, confirm that the following items are present in the figure legend, table legend, main text, or Methods section.

| n/a | Confirmed | |
|---|---|---|
| ☐ | ☒ | The exact sample size (*n*) for each experimental group/condition, given as a discrete number and unit of measurement |
| ☐ | ☒ | A statement on whether measurements were taken from distinct samples or whether the same sample was measured repeatedly |
| ☐ | ☒ | The statistical test(s) used AND whether they are one- or two-sided<br>*Only common tests should be described solely by name; describe more complex techniques in the Methods section.* |
| ☒ | ☐ | A description of all covariates tested |
| ☐ | ☒ | A description of any assumptions or corrections, such as tests of normality and adjustment for multiple comparisons |
| ☐ | ☒ | A full description of the statistical parameters including central tendency (e.g. means) or other basic estimates (e.g. regression coefficient) AND variation (e.g. standard deviation) or associated estimates of uncertainty (e.g. confidence intervals) |
| ☐ | ☒ | For null hypothesis testing, the test statistic (e.g. *F*, *t*, *r*) with confidence intervals, effect sizes, degrees of freedom and *P* value noted<br>*Give P values as exact values whenever suitable.* |
| ☒ | ☐ | For Bayesian analysis, information on the choice of priors and Markov chain Monte Carlo settings |
| ☒ | ☐ | For hierarchical and complex designs, identification of the appropriate level for tests and full reporting of outcomes |
| ☒ | ☐ | Estimates of effect sizes (e.g. Cohen's *d*, Pearson's *r*), indicating how they were calculated |

*Our web collection on statistics for biologists contains articles on many of the points above.*

## Software and code

Policy information about availability of computer code

| Data collection | SpectroFlo Software - Cytek Aurora (Cytek)<br>AID ELISpot software version 7.0 (AID Autoimmun Diagnostika GmbH) - AID Classic ELISpot Reader |
|---|---|
| Data analysis | FlowJo Software (10.6.1, FlowJo LLC, BD Life Sciences), Affinity designer,<br>R version 4.0.1 (R Core Team 2020):<br>R studio 1.3.959, dplyr, FlowSOM, flowStats, ggplot2, Hmisc, pheatmap, Stats, UMAP, flowcore |

For manuscripts utilizing custom algorithms or software that are central to the research but not yet described in published literature, software must be made available to editors and reviewers. We strongly encourage code deposition in a community repository (e.g. GitHub). See the Nature Portfolio guidelines for submitting code & software for further information.

## Data

Policy information about <u>availability of data</u>

All manuscripts must include a <u>data availability statement</u>. This statement should provide the following information, where applicable:

- Accession codes, unique identifiers, or web links for publicly available datasets
- A description of any restrictions on data availability
- For clinical datasets or third party data, please ensure that the statement adheres to our <u>policy</u>

This study did not generate new reagents. Data are available in a public repository (https://doi.org/10.5281/zenodo.7734088). This study did not generate new codes. The codes that support these findings have been previously described[54–63].

## Human research participants

Policy information about <u>studies involving human research participants and Sex and Gender in Research.</u>

| | |
|---|---|
| Reporting on sex and gender | Data relating to the gender was collected on the open Phase IIB clinical trial - ECEHeVac, NCT04988048 |
| Population characteristics | 497 volunteers (age range 18-82 years old) were enrolled in a randomized, open Phase IIB clinical trial (ECEHeVac, NCT04988048) aimed at comparing the immunogenicity and reactogenicity of heterologous and homologous vaccination regimens available in Córdoba, Argentina. |
| Recruitment | Eligible participants were healthy volunteers older than 18 years who had received a first dose of the AZD, BBIBP, Sput 26, or mRNA-1273 vaccine 30-120 days prior to the enrolment date. Exclusion criteria were: immunocompromised status with underlying disease or immunosuppressive treatment; pregnancy and lactation; having received a major surgical intervention in the 30 days prior to the enrolment date; having had a severe allergic reaction (anaphylaxis) to any vaccine; having a visceral disease that lead to disability (heart failure, kidney failure, respiratory failure, liver failure, intestinal malformations, electro-dependence, or having had a visceral transplant less than 2 years previously); and having had COVID-19 (symptomatic or asymptomatic) or a positive anti-nucleocapsid IgG via ELISA on T1 (except for those subjects that had been vaccinated with BBIBP as the first dose).<br><br>Randomization was performed centrally at the Epidemiology Area of the Ministry of Health of the Province of Córdoba by assigning codes to the participants at the time of their registration, anonymizing their personal information to avoid possible biases.<br><br>Randomization methodology: A list was prepared with participants who met inclusion criteria and did not present exclusion criteria. Randomization was performed with a equal group allocation using random permuted block stratification. Randomization was stratified by age for the groups of 18 to 59 years or 60 and over, and according to the time since the application of the first dose of vaccine (0 to 30 days and 30 to 60 days). |
| Ethics oversight | The study received ethical approval by the Registro Provincial de Investigación en Salud (Provincial Registry of Health Research, REPIS-Cba #4371) |

Note that full information on the approval of the study protocol must also be provided in the manuscript.

# Field-specific reporting

Please select the one below that is the best fit for your research. If you are not sure, read the appropriate sections before making your selection.

☒ Life sciences ☐ Behavioural & social sciences ☐ Ecological, evolutionary & environmental sciences

For a reference copy of the document with all sections, see nature.com/documents/nr-reporting-summary-flat.pdf

# Life sciences study design

All studies must disclose on these points even when the disclosure is negative.

| | |
|---|---|
| Sample size | No sample size calculation was performed. We used all the samples available in each dataset. |
| Data exclusions | For FACS, samples with fewer than 1000 live cells were excluded. For ELISpot, samples were excluded if the negative control wells had more than 39 or the positive control wells fewer than 40 spots. |
| Replication | All experiments have multiple replicates. All results were performed in at least 2 independent experiments. |
| Randomization | Participants were randomized with equal group allocation to determine the vaccine used as Dose 2 |
| Blinding | All investigators and collaborators were blinded to clinical results when performing measurements and assays. |

# Reporting for specific materials, systems and methods

We require information from authors about some types of materials, experimental systems and methods used in many studies. Here, indicate whether each material, system or method listed is relevant to your study. If you are not sure if a list item applies to your research, read the appropriate section before selecting a response.

## Materials & experimental systems

| n/a | Involved in the study |
|---|---|
| ☐ | ☒ Antibodies |
| ☐ | ☒ Eukaryotic cell lines |
| ☒ | ☐ Palaeontology and archaeology |
| ☒ | ☐ Animals and other organisms |
| ☒ | ☐ Clinical data |
| ☒ | ☐ Dual use research of concern |

## Methods

| n/a | Involved in the study |
|---|---|
| ☒ | ☐ ChIP-seq |
| ☐ | ☒ Flow cytometry |
| ☒ | ☐ MRI-based neuroimaging |

## Antibodies

| | |
|---|---|
| Antibodies used | The antibodies are listed in Extended data table 2 and Extended data table 3. |
| Validation | All the antibodies have been validated by the manufacturer and then titrated in house (human PBMCs). Please see clones, fluorochromes and company webpages for specific validation. |

## Eukaryotic cell lines

Policy information about cell lines and Sex and Gender in Research

| | |
|---|---|
| Cell line source(s) | Vero 76 cells (ATCC CRL-1587) |
| Authentication | No authentication for the commercially available cell line. |
| Mycoplasma contamination | Vero 76 cell line was tested negative for mycoplasma contamination. |
| Commonly misidentified lines (See ICLAC register) | No commonly misidentified cell lines were used |

## Flow Cytometry

### Plots

Confirm that:

☒ The axis labels state the marker and fluorochrome used (e.g. CD4-FITC).

☒ The axis scales are clearly visible. Include numbers along axes only for bottom left plot of group (a 'group' is an analysis of identical markers).

☒ All plots are contour plots with outliers or pseudocolor plots.

☒ A numerical value for number of cells or percentage (with statistics) is provided.

### Methodology

| | |
|---|---|
| Sample preparation | Cryopreserved PBMCs were stored in liquid nitrogen. Then, for spectral flow analysis and ELISpot, cells were thawed using Cryo thaw devices (Medax). PBMCs were resuspended in cell culture medium supplemented with 2U/ml benzonase by centrifugation (300 r.c.f.; 7 min; 24C). Cell count was calculated using an automated cell counter (Bio-Rad).For the spike-binding mBC and T cell panels, $1.5 \times 10^6$ and $1.0 \times 10^6$ PBMCs respectively were washed with PBS and blocked using Human TruStain FcX and True-Stain Monocyte Blocker (BioLegend). |
| Instrument | Cytek Aurora (Cytek) |
| Software | SpectroFlo Software |
| Cell population abundance | Expressed as a frequency of the selected population |

Gating strategy

Gating strategy is provided in Extended data 7. For high-dimensional flow cytometry analysis, dead cells, doublets, or cells stained by fluorochrome aggregates were excluded from the analysis via manual gating using FlowJo. Datasets of different batches were corrected using the CytoNorm R package

☒ Tick this box to confirm that a figure exemplifying the gating strategy is provided in the Supplementary Information.

