## [Peer Review File · Nature Immunology]

Peer Review Information

Journal: Nature Immunology

Manuscript Title: High-dimensional analysis of sixteen SARS-CoV-2 vaccine combinations reveals lymphocyte signatures correlating with immunogenicity

Corresponding author name(s): Professor Burkhard Becher, Dr Nicolas Nunez, Dr Mariana Maccioni

Reviewer Comments & Decisions:

Decision Letter, initial version:
--

10th Nov 2022

Dear Burkhard,

Thank you for supplying your point-by-point response to the referees' comments on your manuscript entitled "Systemic lymphocyte signatures dictate the magnitude of immune responses to vaccination". As noted earlier, while they find your work of considerable potential interest, they have raised quite substantial concerns that must be addressed. In light of these comments, we cannot accept the current manuscript for publication, but would be very interested in considering a revised version that addresses these concerns as proposed in your author response.

It seems like much of the response to referee #1 are clarifications and reanalysis of datasets that are already in hand. Additionally, referee #2 likewise suggested reanalysis of the current datasets, as well as an analysis of the vaccine-induced IgA responses, if banked blood samples are still available. Hence, we invite you to submit a substantially revised manuscript, however please bear in mind that we will be reluctant to approach the referees again in the absence of major revisions.

When you revise your manuscript taking into account all reviewer and editor comments, please highlight all changes in the manuscript text file in Microsoft Word format.

* Include a "Response to referees" document detailing, point-by-point, how you addressed each referee comment. If no action was taken to address a point, you must provide a compelling argument.

This response will be sent back to the referees along with the revised manuscript.

* If you have not done so already please begin to revise your manuscript so that it conforms to our Article format instructions at <http://www.nature.com/ni/authors/index.html>. Refer also to any guidelines provided in this letter.

The Reporting Summary can be found here:

When submitting the revised version of your manuscript, please pay close attention to our [href="https://www.nature.com/nature-portfolio/editorial-policies/image-integrity">Digital Image Integrity Guidelines. and to the following points below:](https://www.nature.com/nature-portfolio/editorial-policies/image-integrity)

[REDACTED]

If you wish to submit a suitably revised manuscript we would hope to receive it within 6 months. If you cannot send it within this time, please let us know. We will be happy to consider your revision so long as nothing similar has been accepted for publication at Nature Immunology or published elsewhere.

Nature Immunology is committed to improving transparency in authorship. As part of our efforts in this direction, we are now requesting that all authors identified as 'corresponding author' on published papers create and link their Open Researcher and Contributor Identifier (ORCID) with their account on the Manuscript Tracking System (MTS), prior to acceptance. ORCID helps the scientific community achieve unambiguous attribution of all scholarly contributions. You can create and link your ORCID from the home page of the MTS by clicking on 'Modify my Springer Nature account'. For more information please visit please visit

<http://www.springernature.com/orcid>>www.springernature.com/orcid.

Thank you for the opportunity to review your work.

Sincerely,

Laurie A. Dempsey, Ph.D.
Senior Editor
Nature Immunology
l.dempsey@us.nature.com
ORCID: 0000-0002-3304-796X

Referee expertise:

Referee #1: Vaccine-induced immune responses

Referee #2: Multiomic dataset analysis

Reviewers' Comments:

Reviewer #1:

Remarks to the Author:

This study analysed immune responses in individuals receiving 16 different combinations of homologous and heterologous prime-boost COVID-19 vaccine regimens. This is important given the multiple COVID-19 vaccines being deployed worldwide and the likely need for ongoing intermittent boosters in order to maintain immunological protection. The authors show differences in the phenotype of the B cell response according to vaccine combination given which correlates with the strength of the antibody response. They identified lower expression of CD21, CD38 and CXCR5 on spike-binding mBCs as a marker of responding to vaccination. A lack of mBC class-switching following BBIBP vaccination resulted in the lowest antibody responses, but is also suggested to potentially lead to broader immunity against variants. Group differences in T cell responses determined by IFN- γ ELISpot assay were also apparent, with BBIBP as first dose being the strongest inducer of T cell immunity regardless of the dose 2 vaccine used. Clustering analysis identified 5 main signatures unique to specific vaccine combinations.

It's very helpful seeing these comparative Ab, B cell and T cell data since many COVID-19 vaccine trials test only a few vaccine combinations and many different assays are used to analyse these immune parameters in the various published studies making comparisons between studies difficult. The data clearly indicate that certain regimens are superior to others. These data therefore provide important insights into COVID-19 vaccine mechanisms which should help us better understand the role of booster vaccination in the future.

This is an original and important study. The data are high quality and very clearly presented for what is a complicated cohort given the number of vaccine combinations administered. The statistics used are appropriate and the results are of great interest to immunologists and vaccinologists. The references are also appropriate.

In order to garner more reader interest, I suggest that the title should be more descriptive and mention that they tested 16 combinations of homologous and heterologous COVID-19 vaccination regimens

The Abstract could improve to better capture the significance of the findings. The authors conclude that their data suggest that heterologous regimens may be immunologically superior to homologous regimens for non-mRNA vaccines and this should be stated in the abstract.

There are many extended figures which are cited throughout. At times detail provided in the extended figures is referred to but this needs to be described in the text since many people do not want to go through the figures to find out the key take homes. Where this is necessary is provided in my comments below.

Can the authors better explain and justify early in the text the difference in timing for the T1 blood sample according to vaccine regimen? It's still not clear from methods whether the timing was consistent for specific vaccine groups or random across all groups according to when volunteers presented for 2nd dose.

Line 128-9 – can the authors state in the text how many individuals failed to have detectable NAb and from which vaccine regimens?

Line 131 – please state the fold-increase with mRNA and AZD/Ad5

Line 132 – the authors state highest but then cite multiple groups. It can only be highest in one group. Please explain the hierarchy of responses more clearly.

Line 254-5 – It would be helpful if the authors could state here which groups did have this correlation

Line 342 – a brief speculation on the role of CD4-CD8- double negative T cells would be helpful since many people are not aware of this rare subset.

Line 345 – define KLRG1 when first used

Line 396 – were all volunteers tested for anti-N IgG?

Line 399 – there were multiple vaccines so it's not clear what the authors mean by 1:1 for randomisation

Reviewer #2:

Remarks to the Author:

In this manuscript, Nunez, Schmid, and Power et al leverage a unique cohort of 497 patients who

were randomized to undergo one of 16 homologous or heterologous combinations of vaccination against SARS-CoV-2, including traditional viral vector-based approaches (AZD), inactivated full virus (BBIBP), adenovirus-based vaccines (Sputnik and Ad5), protein-based vaccines (Novavax), and mRNA vaccines (mRNA-1273), and then assessed humoral and cellular immunity at 1 and 3 months using a combination

The primary findings of the study are as follows:

1. Of the vaccine combinations, mRNA vaccination, either alone or as a second dose following an alternative first dose, was clearly associated both the highest levels of Spike IgG and neutralizing antibody titers. This correlated well with the frequency of spike-binding memory B-cells at 3 months following the first dose.
2. Phenotypically, they find that mRNA vaccination leads to the highest level of either “switched-activated” or “atypical” memory B-cells, while the lowest of these cell populations was seen in BBIBP-vaccinated patients.
3. Using an IFN γ -ELISPOT assay, they show that BBIBP is the most potent inducer of what are likely to be CD8+ T-cell responses to vaccination, in combination with any heterologous booster. They also show, as expected, that BBIBP was the only vaccine able to induce nucleocapsid-specific T-cell responses.
4. They then interrogated whether differences in antigen-specific ELISPOT responses were associated with broader differences in peripheral blood T-cell phenotypes. They did not observe any significant association of ELISPOT responses with broad naïve/memory immune phenotyping. They then did broader 27-color spectral flow cytometry followed by FlowSOM clustering, which per their extended data table generated 80 cell clusters. Several of these clusters correlated with ELISPOT responses, including cells negative for both CD4 and CD8, some effector memory CD8+ T-cell subsets, and some central memory CD4+ T-cell subsets, which correlated positively with responses, and some effector memory CD4+ T-cell subsets, which correlated negatively with responses, although these correlations are quite weak (all r of 0.2-0.3). They also suggest that there is a correlation of humoral and cellular responses to vaccination across vaccine regimens in Figure 6b, although this correlation is very weak and is contradicted by the rest of the manuscript that suggests that distinct regimens confer distinct combinations of cellular and humoral immune responses.
5. Finally, the authors combine their spike-specific and broad T-cell immunophenotyping with their antibody and ELISPOT data to try to identify immune response ‘signatures’ to vaccination (Figure 6c). They identify 5 main signatures and once again identify the combination of BBIBP and mRNA in providing both robust humoral and CD8+ T-cell responses to vaccination.

The study has some significant advantages, including the impressive sample size and study design with randomization of first and second doses of vaccine that allows for direct comparison of vaccine regimens. Some of the results are quite clear – primarily the fact that mRNA vaccination induces the most robust humoral immune response while inactivated virus produces the strongest CD8+ T-cell response.

The main detriment of the study lies in the mechanistic insights gained from the assays performed, particularly with respect to understanding how specific heterologous vaccinations promote T-cell

immunity. Specifically, the manuscript emphasizes identifying both B-cell and T-cell immunophenotypes that are suggestive of vaccine responses, and this is by a wide margin the weakest data in the manuscript. Because of this, my overall sense is that the primary utility of this study may be more clinical and/or translational than mechanistic and as such it may be more suited for a journal other than Nature Immunology.

My specific comments are as follows:

1. The Spike IgG and neutralizing antibody titer data for the distinct vaccination strategies is far more compelling than the analysis of spike-specific memory B-cells. For example, the association of individual flow markers such as CD21 or CD38 MFI with neutralizing antibody titers is fairly weak across all markers, and association of individual marker MFIs is not typically how flow cytometry data is analyzed. It would make more sense to see how the frequency of B-cells with specific phenotypes (for example see Mathew et al Science 2020 or Kato et al J Infect Dis 2022) correlate with neutralizing antibodies, particularly in the spike-binding population. Even if this is done, however, I'm not sure that the spike-specific memory B-cell phenotyping is adding much beyond what the spike IgG titers and nAb data shows. Is there actually evidence that mRNA-1273 induced more class-switched COVID-specific antibody production? For example, are Spike-specific IgA levels higher?
2. One of the primary issues with using IFN γ -ELISPOT for cellular vaccine responses is that it a) does not allow for assessment of which cells are producing IFN-g and b) is likely to primarily emphasize CD8+ T-cell responses to vaccination, which is problematic given that multiple reports suggest a stronger CD4+ T-cell response to vaccination. This problem also manifests as the fact that the authors are attempting to correlate broad (non-antigen-specific) immunophenotypes with functional responses. Unsurprisingly, these associations are either not present (Extended Figures 6a-c) or quite weak (Figure 4d-f, Figure 5).
3. There are several issues of concern with the T-cell immunophenotyping strategy in Figures 4-6. It is hard to believe based on both the UMAPs and simply on the fact that only 27 parameters are assessed that there could be 80 discrete immunological clusters generated. The authors don't provide any explanation of what distinguishes any of the clusters associated with response (such as EM CD8+ clusters 71 and 72) from those that don't (clusters 13, 58, 59, 60, 14, 20, 22, 23, 31, 33, 39, 4). This is particularly concerning given that the associations of these somewhat arbitrary sub-clusters with IFN-g responses is very weak ($R < 0.5$). To be believable, the authors would have to more directly correlate cellular phenotypes with functional responses using either 1) activation-induced marker (AIM) assays or 2) tetramer or dextramer-specific immunophenotyping.
4. The suggestion in Figure 6b that antibody and cellular T-cell responses are correlated is odd and contradicted by the rest of their manuscript. The associations are extremely weak ($r = 0.24$ and 0.3) and this is essentially confirmed by 6c which seems to suggest that the drivers of T-cell and B-cell responses are distinct and therefore optimized when different vaccine strategies are combined.
5. Finally, the title is a significant overstatement of the conclusions of the study. There is simply no evidence to support that systemic lymphocyte signatures DICTATE the magnitude of response to vaccination in this study. This would be at best an association/correlation.

Author Rebuttal to Initial comments

See Inserted PDF

Dear Laurie,

Thanks for giving us the opportunity to answer your referee's queries. Below, you will find details of how we improved the manuscript according to the suggestions.

Reviewer #1

This study analysed immune responses in individuals receiving 16 different combinations of homologous and heterologous prime-boost COVID-19 vaccine regimens. This is important given the multiple COVID-19 vaccines being deployed worldwide and the likely need for ongoing intermittent boosters in order to maintain immunological protection. The authors show differences in the phenotype of the B cell response according to vaccine combination given which correlates with the strength of the antibody response. They identified lower expression of CD21, CD38 and CXCR5 on spike-binding mBCs as a marker of responding to vaccination. A lack of mBC class-switching following BBIBP vaccination resulted in the lowest antibody responses, but is also suggested to potentially lead to broader immunity against variants. Group differences in T cell responses determined by IFN- γ ELISpot assay were also apparent, with BBIBP as first dose being the strongest inducer of T cell immunity regardless of the dose 2 vaccine used. Clustering analysis identified 5 main signatures unique to specific vaccine combinations.

It's very helpful seeing these comparative Ab, B cell and T cell data since many COVID-19 vaccine trials test only a few vaccine combinations and many different assays are used to analyse these immune parameters in the various published studies making comparisons between studies difficult. The data clearly indicate that certain regimens are superior to others. These data therefore provide important insights into COVID-19 vaccine mechanisms which should help us better understand the role of booster vaccination in the future.

This is an original and important study. The data are high quality and very clearly presented for what is a complicated cohort given the number of vaccine combinations administered. The statistics used are appropriate and the results are of great interest to immunologists and vaccinologists. The references are also appropriate.

Authors' response: We thank the reviewer for the positive feedback and the astute summary of our work.

In order to garner more reader interest, I suggest that the title should be more descriptive and mention that they tested 16 combinations of homologous and heterologous COVID-19 vaccination regimens

Authors' response: We agree and changed the title to: *High-dimensional analysis of sixteen SARS-CoV-2 vaccine combinations reveals lymphocyte signatures correlating with immunogenicity.*

The Abstract could improve to better capture the significance of the findings. The authors conclude that their data suggest that heterologous regimens may be immunologically

superior to homologous regimens for non-mRNA vaccines and this should be stated in the abstract.

Authors' response: We thank Referee #1; we agree, rewrote the abstract and added: "For non-mRNA vaccines, *heterologous combinations were generally more immunogenic in terms of specific antibodies and cellular responses compared to homologous regimens*".

There are many extended figures which are cited throughout. At times detail provided in the extended figures is referred to but this needs to be described in the text since many people do not want to go through the figures to find out the key take homes. Where this is necessary is provided in my comments below.

Can the authors better explain and justify early in the text the difference in timing for the T1 blood sample according to vaccine regimen? It's still not clear from methods whether the timing was consistent for specific vaccine groups or random across all groups according to when volunteers presented for 2nd dose.

Authors' response: We agree with the reviewer and have adapted the corresponding results and methods sections in the manuscript on lines 108-113.

Line 128-9 – can the authors state in the text how many individuals failed to have detectable NAb and from which vaccine regimens?

Authors' response: Excellent idea; we have added the number of volunteers without detectable NAb to the results section in the manuscript on lines 193-198 (also see **Figure 3e**).

Line 131 – please state the fold-increase with mRNA and AZD/Ad5

Authors' response: We added this information on lines 139-144 of the revised version of the manuscript.

Line 132 – the authors state highest but then cite multiple groups. It can only be highest in one group. Please explain the hierarchy of responses more clearly.

Authors' response: We thank Referee #1 for pointing this out; we edited the corresponding text in lines 136-139 to better explain the hierarchy of responses

Line 254-5 – It would be helpful if the authors could state here which groups did have this correlation

Authors' response: In addition to Extended Data Fig. 7a, we updated the manuscript to mention the correlating group in the text on lines 265-266.

Line 342 – a brief speculation on the role of CD4-CD8- double negative T cells would be helpful since many people are not aware of this rare subset.

Authors' response: We agree with Reviewer #1 and expanded the discussion of CD4-CD8- double negative T cells on lines 362-365.

Line 345 – define KLRG1 when first used

Authors' response: Accordingly, we have added a sentence to the discussion on lines 358-359.

Line 396 – were all volunteers tested for anti-N IgG?

Authors' response: Yes, all volunteers were tested for anti-N IgG. We made this point clearer in the methods section on lines 411-413.

Line 399 – there were multiple vaccines so it's not clear what the authors mean by 1:1 for randomisation

Authors' response: We have updated in Material and Methods on lines 415-416.

Reviewer #2

(Remarks to the Author)

In this manuscript, Nunez, Schmid, and Power et al leverage a unique cohort of 497 patients who were randomized to undergo one of 16 homologous or heterologous combinations of vaccination against SARS-CoV-2, including traditional viral vector-based approaches (AZD), inactivated full virus (BBIBP), adenovirus-based vaccines (Sputnik and Ad5), protein-based vaccines (Novavax), and mRNA vaccines (mRNA-1273), and then assessed humoral and cellular immunity at 1 and 3 months using a combination

The primary findings of the study are as follows:

1. Of the vaccine combinations, mRNA vaccination, either alone or as a second dose following an alternative first dose, was clearly associated both the highest levels of Spike IgG and neutralizing antibody titers. This correlated well with the frequency of spike-binding memory B-cells at 3 months following the first dose.
2. Phenotypically, they find that mRNA vaccination leads to the highest level of either “switched-activated” or “atypical” memory B-cells, while the lowest of these cell populations was seen in BBIBP-vaccinated patients.
3. Using an IFNg-ELISPOT assay, they show that BBIBP is the most potent inducer of what are likely to be CD8+ T-cell responses to vaccination, in combination with any heterologous booster. They also show, as expected, that BBIBP was the only vaccine able to induce nucleocapsid-specific T-cell responses.

4. They then interrogated whether differences in antigen-specific ELISPOT responses were associated with broader differences in peripheral blood T-cell phenotypes. They did not observe any significant association of ELISPOT responses with broad naïve/memory immune phenotyping. They then did broader 27-color spectral flow cytometry followed by FlowSOM clustering, which per their extended data table generated 80 cell clusters. Several of these clusters correlated with ELISPOT responses, including cells negative for both CD4 and CD8, some effector memory CD8⁺ T-cell subsets, and some central memory CD4⁺ T-cell subsets, which correlated positively with responses, and some effector memory CD4⁺ T-cell subsets, which correlated negatively with responses, although these correlations are quite weak (all r of 0.2-0.3). They also suggest that there is a correlation of humoral and cellular responses to vaccination across vaccine regimens in Figure 6b, although this correlation is very weak and is contradicted by the rest of the manuscript that suggests that distinct regimens confer distinct combinations of cellular and humoral immune responses.

5. Finally, the authors combine their spike-specific and broad T-cell immunophenotyping with their antibody and ELISPOT data to try to identify immune response 'signatures' to vaccination (Figure 6c). They identify 5 main signatures and once again identify the combination of BBIBP and mRNA in providing both robust humoral and CD8⁺ T-cell responses to vaccination.

The study has some significant advantages, including the impressive sample size and study design with randomization of first and second doses of vaccine that allows for direct comparison of vaccine regimens. Some of the results are quite clear – primarily the fact that mRNA vaccination induces the most robust humoral immune response while inactivated virus produces the strongest CD8⁺ T-cell response.

Authors' response: We thank the reviewer for the positive feedback and summary of our work. We would like to point out, however, that Novavax was not part of the analysis. Equally, the time points were not 1 and 3 months, but instead the first time point (T1) was 4-12 weeks after the first vaccine dose, T2 was two weeks after the second dose, and T3 was four weeks after the second dose.

The main detriment of the study lies in the mechanistic insights gained from the assays performed, particularly with respect to understanding how specific heterologous vaccinations promote T-cell immunity. Specifically, the manuscript emphasizes identifying both B-cell and T-cell immunophenotypes that are suggestive of vaccine responses, and this is by a wide margin the weakest data in the manuscript. Because of this, my overall sense is that the primary utility of this study may be more clinical and/or translational than mechanistic and as such it may be more suited for a journal other than Nature Immunology.

Authors' response: We agree that the applied methods are not suitable to fully interrogate the mechanistic underpinnings of specific T cell responses to vaccination (for this, we believe a preclinical model system is required). However, the main aim of this study was to identify - for the first time - the phenotypic signatures in the B and T cell compartments after vaccination

in the context of different types of vaccines in a single-cohort. This complex immunophenotype map was then linked to the antibody and functional cellular responses.

Our results reveal the B cell phenotypes and T cell clusters that could be used as biomarkers for testing immune responses to newly developed vaccines and also pave the way for further investigations into the exact mechanisms behind the associations between the cellular and antibody responses. Also, to our knowledge, this is the largest single center side-by-side vaccine comparison to date, where we have the range of antibody data, spike-specific responses and comprehensive immune profiles across 16 vaccine combinations. We remain confident that the cellular responses with regards to phenotype are solid indicators of the vaccine responses across individuals. We agree that indeed some of the differences in individual marker expression and/or changes in subset frequencies are moderate, albeit statistically significant.

To further emphasize the true power of our multi-omics approach, we have selected the top features and visualized them in a PCA, shown in **new Fig. 6d** in the revised manuscript. This shows a clear separation of the vaccine regimens driven by the identified immune signatures. We believe that this PCA elegantly showcases that a combination of complex immune features and functional readouts can provide a deep understanding of the interplay of vaccine regimens and immunity. We trust that this provides Reviewer #2 with additional confidence in the data.

My specific comments are as follows:

1. The Spike IgG and neutralizing antibody titer data for the distinct vaccination strategies is far more compelling than the analysis of spike-specific memory B-cells. For example, the association of individual flow markers such as CD21 or CD38 MFI with neutralizing antibody titers is fairly weak across all markers, and association of individual marker MFIs is not typically how flow cytometry data is analyzed. It would make more sense to see how the frequency of B-cells with specific phenotypes (for example see Mathew et al Science 2020 or Kato et al J Infect Dis 2022) correlate with neutralizing antibodies, particularly in the spike-binding population. Even if this is done, however, I'm not sure that the spike-specific memory B-cell phenotyping is adding much beyond what the spike IgG titers and nAb data shows. Is there actually evidence that mRNA-1273 induced more class-switched COVID-specific antibody production? For example, are Spike-specific IgA levels higher?

Authors' response: We thank Reviewer #2 for the comment and for pointing out the importance of the antibody titer data. While we agree that every single association individually showed moderate Spearman coefficients, the p-values were significant and in combination the patterns were striking (see **new Fig. 6d**). We agree that comparing the MFI of individual markers is not the most typical method to analyze flow cytometry data, but it represents a valid option particularly for high-parametric data, since biologically, some markers are expressed on a continuum rather than simply yes or no (as also done in e.g. Turner et al., Nature, 2020 (Extended Data Fig. 2); Ellebedy et al., Nature Immunology, 2016 (Fig. 1); or Zhang et al., Cell, 2022 (Fig. S6E).

As requested, for reviewer clarification, we have analyzed and correlated the frequencies of spike-specific memory B cells for the observed phenotypes with antibody response. However, across all the parameters analyzed, we observed no significant differences (NS p-value; see example below). This analysis confirms that MFI shows the main differences on the analyzed groups.

a Relative expression by Spike-binding mBCs (T3)

b Spike-binding mBCs (T3)

Phenotype of spike-binding mBCs from participants receiving different vaccine regimens. (a) The scaled frequency of phenotypic markers by spike-binding mBCs at T3 for each group. (b) Correlations between phenotypic marker frequency by spike-binding mBCs and antibody response (anti-S-RBD IgG levels and NAb titres) across all vaccine regimens at T3. Colour indicates the Spearman's rank correlation coefficient (r_s), and the bubble size indicates the p-value. (c) Participants were classified as negative (< 10) or positive (≥ 10) for NAb titres and marker frequency levels on spike-binding mBC were compared between negative and positive participants. (d) Participants were classified as negative (< 1.03 , the detection threshold) or positive (> 1.03) for IFN γ response and marker frequency levels on spike-binding mBC were compared between negative and positive participants. Boxes bound the interquartile range (IQR) divided by the median, and Tukey-style whiskers extend to a maximum of $1.5 \times \text{IQR}$ beyond the box. Dots are participant data points. P-values for (a,c-d) were calculated using the Mann-Whitney-Wilcoxon test and corrected for multiple hypothesis testing with the Benjamini-Hochberg method. Only statistically significant p-values ($p < 0.05$) are displayed.

Since the phenotypes of specific memory B cells have not yet been compared between so many different vaccine platforms in a single human cohort, we remain confident that our results are novel and represent a great conceptual advance in that different types of vaccines result in phenotypically different antigen-specific memory B cells. The correlation with antibody levels shows the relevance of this phenotype for the antibody response and thus introduces a concept relevant for future vaccine development.

The reviewer suggested to analyze the Anti-S-RBD IgA in our cohort. We felt that this was an excellent suggestion. We analyzed anti-S-RBD IgA across all 349 donors at T3 and included these new results in the revised version of the manuscript (**new Figure 2e/3b/6a-d** and **new Extended Figure 2d/4c-d/7a**). Finally, we adapted the results and discussion of the antibody response as well as the discussion of differences in class-switching accordingly.

2. One of the primary issues with using IFN γ -ELISPOT for cellular vaccine responses is that it a) does not allow for assessment of which cells are producing IFN-g and b) is likely to primarily emphasize CD8 $^+$ T-cell responses to vaccination, which is problematic given that multiple reports suggest a stronger CD4 $^+$ T-cell response to vaccination. This problem also manifests as the fact that the authors are attempting to correlate broad (non-antigen-specific) immunophenotypes with functional responses. Unsurprisingly, these associations are either not present (Extended Figures 6a-c) or quite weak (Figure 4d-f, Figure 5).

Authors' response: We thank Reviewer #2 for making this important point. Indeed, with this method we cannot further phenotype the cells that are producing IFN-g, as we had discussed in the manuscript (Line 366). Nonetheless, ELISpot has been used in multiple important vaccine trials to determine the IFN γ response (Folegatti, 2020, Lancet; Munro, 2021, Lancet; Zhu, 2020, Lancet; Chen, 2022, Lancet Microbe; Collier, 2021, JAMA; Kalimuddin, 2021, Med; Huat Khoo, 2022, Med) and has been correlated with protection (Menges, 2022, Nat Comm). Also, we chose this method because it has a much higher sensitivity for antigen-specific cells compared to AIM assays or tetramer/dextramer-specific immunophenotyping (Anthony, 2003, Methods, Barabas 2017 BMC Immunology). Furthermore, stimulation with overlapping 15 amino acids long SARS-CoV-2 peptides has been shown to efficiently stimulate both CD4 $^+$ and CD8 $^+$ T cells (i.e. Pozzetto, 2021, Nature; Schmidt, 2021, Nature Medicine; Barros-Martins, 2021; Nature Medicine; Ewer, 2020, Nature Medicine). The rationale behind correlating non-antigen specific immunophenotypes with functional responses is not to deeply phenotype the antigen-specific T cells (which is not possible with an ELISpot assay). Instead, we wanted to compare the strength of the antigen-specific T cell response after vaccination and to find cell subsets that serve as indicators for a potent vaccine response. Finally, while the single measured associations indeed cannot precisely characterize a vaccine response alone, we show in the updated Fig. 6 in the revised manuscript that combining a limited subset of immunological features can be used to clearly discriminate groups of vaccine regimens.

3. There are several issues of concern with the T-cell immunophenotyping strategy in Figures 4-6. It is hard to believe based on both the UMAPs and simply on the fact that only 27 parameters are assessed that there could be 80 discrete immunological clusters generated. The authors don't provide any explanation of what distinguishes any of the clusters associated with response (such as EM CD8 $^+$ clusters 71 and 72) from those that

don't (clusters 13, 58, 59, 60, 14, 20, 22, 23, 31, 33, 39, 4). This is particularly concerning given that the associations of these somewhat arbitrary sub-clusters with IFN-g responses is very weak ($R < 0.5$). To be believable, the authors would have to more directly correlate cellular phenotypes with functional responses using either 1) activation-induced marker (AIM) assays or 2) tetramer or dextramer-specific immunophenotyping.

Authors' response: We would like to clarify that we applied FlowSOM to generate the 80 clusters, which is a clustering algorithm that has been validated for the analysis of cytometry data. This constitutes an unbiased approach, where we need to be agnostic and not be influenced by knowledge regarding canonical immune features (such as expansion of TCM cells). The advantage of this method is that it allows us to identify infrequent and specialized cell subsets that might otherwise be hidden in the millions of cells analyzed. Based on the cellular complexity, FlowSOM clustering allows the user to select the number of clusters to identify, and even provides a default of 100 clusters in order to provide high resolution of the cell types. Eighty, 100, or more clusters is appropriate and not at all unusual for high dimensional flow cytometry with panels of 27 markers (Quintelier et al. 2021 Nature Protocols; Roukens et al. 2021 Nature Immunol). In theory, 27 markers result in more than 134 million dimensions. Therefore, the clustering was not arbitrary, but strikes a balance between data resolution and biological interpretability. Finally, we did indeed merge some of the 80 clusters based on their similarity to each other, resulting in fewer than 80 defined clusters. While we appreciate that the use of AIM assays or multimers would provide some additional insight into the nature of SARS-CoV-2-reactive T cells, the combination of analyzing the overall immunophenotype with single-cell resolution and ultrasensitive functional readouts (ELISpot) was able to clearly separate the vaccine regimens (see new **Fig. 6d**).

We regret that the T cell clusters identified were not clearly annotated in the manuscript. In the revised version we provide a heatmap (**new Extended data table 3**) showing the phenotypes of the correlating and non-correlating clusters.

4. The suggestion in Figure 6b that antibody and cellular T-cell responses are correlated is odd and contradicted by the rest of their manuscript. The associations are extremely weak ($r = 0.24$ and 0.3) and this is essentially confirmed by 6c which seems to suggest that the drivers of T-cell and B-cell responses are distinct and therefore optimized when different vaccine strategies are combined.

Authors' response: We thank Reviewer #2 for the comment. We fully agree. In order to clarify the key findings, we elected to remove the Figure 6b that did not indicate the final vaccine groups, resulting in confusion. **Fig. 6b** was updated in the revised manuscript to emphasize the main findings of our work.

5. Finally, the title is a significant overstatement of the conclusions of the study. There is simply no evidence to support that systemic lymphocyte signatures DICTATE the magnitude of response to vaccination in this study. This would be at best an association/correlation.

Authors' response: We thank the Reviewer for raising this point. We agree and as suggested by the reviewers we updated the title to: *High-dimensional analysis of sixteen SARS-CoV-2 vaccine combinations reveals lymphocyte signatures correlating with immunogenicity.*

Decision Letter, first revision:

15th Feb 2023

Dear Burkhard,

Thank you for submitting your revised manuscript entitled, "High-dimensional analysis of sixteen SARS-CoV-2 vaccine combinations reveals lymphocyte signatures correlating with immunogenicity" has now been seen by 2 referees (one original and a new referee, as we were unable to obtain the other previous reviewer). Overall the manuscript is much improved, but the referees have raised some important points that can likely be addressed by providing clarifications in the text. Thus, we are interested in the possibility of publishing your study in Nature Immunology, if revised to address the remaining comments of the referees.

We therefore invite you to revise your manuscript taking into account all reviewer and editor comments. Please highlight all changes in the manuscript text file in Microsoft Word format.

- * Include a "Response to referees" document detailing, point-by-point, how you addressed each referee comment. If no action was taken to address a point, you must provide a compelling argument. This response will be sent back to the referees along with the revised manuscript.
- * If you have not done so already please begin to revise your manuscript so that it conforms to our Article format instructions at <http://www.nature.com/ni/authors/index.html>. Refer also to any guidelines provided in this letter.
- * Please include a revised version of any required reporting checklist. It will be available to referees to aid in their evaluation of the manuscript goes back for peer review. They are available here:

Reporting summary:

When submitting the revised version of your manuscript, please pay close attention to our [href="https://www.nature.com/nature-portfolio/editorial-policies/image-integrity">Digital Image Integrity Guidelines. and to the following points below:](https://www.nature.com/nature-portfolio/editorial-policies/image-integrity)

Finally, please ensure that you retain unprocessed data and metadata files after publication, ideally

archiving data in perpetuity, as these may be requested during the peer review and production process or after publication if any issues arise.

[REDACTED]

We hope to receive your revised manuscript within four weeks. If you cannot send it within this time, please let us know. We will be happy to consider your revision so long as nothing similar has been accepted for publication at Nature Immunology or published elsewhere.

Nature Immunology is committed to improving transparency in authorship. As part of our efforts in this direction, we are now requesting that all authors identified as 'corresponding author' on published papers create and link their Open Researcher and Contributor Identifier (ORCID) with their account on the Manuscript Tracking System (MTS), prior to acceptance. ORCID helps the scientific community achieve unambiguous attribution of all scholarly contributions. You can create and link your ORCID from the home page of the MTS by clicking on 'Modify my Springer Nature account'. For more information please visit www.springernature.com/orcid.

Kind regards,

Laurie

Laurie A. Dempsey, Ph.D.
Senior Editor
Nature Immunology
l.dempsey@us.nature.com
ORCID: 0000-0002-3304-796X

Reviewers' Comments:

Reviewer #1:

Remarks to the Author:

The authors have addressed my questions and suggestions but there are still some minor aspects that

need further editing.

The abstract needs further work to capture the key findings e.g. in particular there is no mention of the superiority of mRNA vaccines across a number of measured outcomes. Also priming with BBIP induced higher Abs than what?

Line 277 It is a bit simplistic to describe these mBCs as representing an activated phenotype.

Line 342-3 It's not clear why dengue is mentioned here and how it relates to the conclusions being drawn.

Line 351-3 I don't follow the authors thinking with this speculation. What cytokines specifically are they talking about and what proof do they have?

Line 357-8 The T cells expressing high CD27 and CD45RA are therefore naïve T cells so this should be stated

Lines 364-5 Stating they "are involved in the immune response against viruses and other pathogens" is a very vague description of the role of DN cells and gives no indication of what they do. Given 2 CD4-/CD8- clusters are expanded they warrant further discussion.

Line 373 Maybe best to say "better protection" rather than "reliable protection".

Line 443 "collected at T3" not "collected on T3"

Reviewer #3:

Remarks to the Author:

The authors elegantly study humoral and cellular responses to 16 homologous and heterologous prime boost vaccination schedules for SARS-CoV-2 encompassing a variety of vaccine technologies. This is a very useful study reflecting world experiencing in response to COVID-19 which has required adaptation of vaccine schedules based on availability.

Methodically characterising the immunological landscape across multiple vaccine combinations is highly informative for future vaccine design as previous vaccine studies have covered only limited combinations and comparisons of results are problematic.

The authors combine data on antibody responses, antigen specific B cell responses, T cell functional responses to SARS-CoV-2 peptides and global T cell subset frequencies to define four immune signatures associated with the varying SARS-CoV-2 vaccine combinations.

Appropriate methods and analysis have been used and the data is well presented, with a coherent description of the complex analysis of multiple subgroups. The discussion appropriately addresses the limitations of the study. As the authors conclude, this study has implications for considering which might be the most appropriate vaccine technologies and schedules for other pathogens. Subsequent correlation of these results with protection from infection is the key to harnessing these results.

Suggestions and clarifications:

I have some comments regarding the clarity of the report and have highlighted points at which I think additional information would strengthen the manuscript. There is a wealth of data to display and at times it is difficult to ascertain the most pertinent results, especially with the number of extended figures. In specific instances detailed below I felt that key results are not reported in the text / figures at an early enough point and come much later than I was expecting.

Line 79: Novavax is mentioned in the introduction as an example of a recombinant spike nanoparticle vaccine. Whilst it is useful to highlight the other technologies available it should perhaps be made

clearer that this is not a vaccine technology being assessed in the current study.

Study design section: Whilst a broad range of ages are covered in this cohort, within the subgroups receiving the different vaccine regimens it is evident that there is some disparity. The mean age of some of the groups is higher than others, at the most extreme 54.7 for Sput26 followed by BBIBP, vs 26.2 for BBIBP followed by AZ, a difference of almost 30 years – are these significant difference? Given the age related changes in the immune response, including those observed to SARS-CoV-2 vaccination (Collier et al., Nature 2021; Ferrerira et al., medrxiv 2022) can the authors comment on this / address this potential limitation? This is especially relevant given the timing of this study in relation to recommendations about use of the AZ vaccine with concerns regarding thrombotic events – it would be useful to comment if any age related restrictions were imposed on the use of AZ vaccines at the time of this study in Argentina.

The serological results are a key finding in terms of potential correlates of protection, initially only feature in the extended figures until comparisons with cellular components are analysed. Is it feasible to move some of these into the main figures? I note the addition of Figure 3e following the comments from Reviewer 1, and this is very important, however, I am not sure about its placement. I feel it would sit better with the data in the 'strength of antibody response varies between vaccines' section and had in fact made a note of its absence here, only to discover it was then addressed in the B cell phenotyping section. If at all possible I think it would be useful to move this earlier in the results. Line 132 should this read anti-S RBD IgG and anti-S RBD IgA as mentioned in the preceding sentence?

Lines 133-135 Why are only IgG and nAb results (Extended data Fig 2a,b) and not IgA featured here? What about the longitudinal IgA response following vaccination? Were they detectable in all and did they increase as with IgG? Additionally, there is no statement of the number of individuals who achieved detectable titres or whether a boosting effect was observed as seen with IgG and nAb (extended data Fig 3 a,b). Could these results be detailed in the text even if editorial restrictions mean it is not possible to show all of these data?

Lines 146-149: The highest median values of IgG to RBD and nAb are observed with the homologous mRNA vaccine schedule, not with a heterologous schedule, but there is limited fold change as demonstrated by the high values already present at T1 after priming with this schedule. As these data are currently in the extended figures as opposed to the main body I think it may be worth explicitly stating this finding or making this text a little clearer.

Line 153 Ref 37 – I think that other references may be more appropriate here with more B cell focus than this review.

Line 156 – How are the authors defining spike specific memory B cells (which markers are used to define as being a MBC)? I could not see this detailed in the methods or the figure legend. Is it the same gating strategy as detailed in the supplementary data of the reference? Are the authors able to show example plots of the spike probe staining and did they use decoy probes or dual staining to help demonstrate the specificity of the staining?

Lines 167-171 I am not entirely sure what the authors are suggesting here – the spike MBC % correlates with serological results (as previously demonstrated in the literature), 'this could serve as an indicator for the systemic antibody response in certain cases'. I initially read this to mean the

authors are suggesting that cellular phenotyping could be used in lieu of serological analysis, which on a purely practical level I'm not sure would be of any use! Apologies if I have misunderstood this statement.

I am concerned that the T cell ELISpot responses are somewhat overstated. Could the authors state the frequency of individuals with IFN- γ responses above the threshold at T1 and T3 shown in Fig5d earlier in the results as this seemed notably absent during the initial reading of the T cell responses section? Figure 4a and c demonstrate that a proportion of individuals (what seems a high proportion in some of the subgroups) do not have responses above the threshold that the authors have defined for detection, and the subsequent comparison of these results to show a significant increase in the response when there is no above threshold response for a proportion of the individuals analysed feels a little misleading. One option is to demonstrate those who go from undetectable to detectable responses perhaps? Additionally, the fold increase observed in the AZD/mRNA-1273 group seems to be largely driven by one extreme outlier – whilst I appreciate this is of course real world data it is important not to overstate this finding, and the BBICP primed groups seem much more convincing.

Line 229 – were any responses observed nucleocapsid peptides in the solely spike vaccinated individuals? 'Strong' seems an inappropriate word to use in this sentence, and it should be clarified if there were any responses to N protein observed in response to spike vaccination – as the authors state this would not be expected.

Line 255-258: this summary of the findings from IFN- γ positive responders is somewhat confusing for the total cohort, given the subsequent description of the differing immune signatures. IFN- γ positive responders have hallmarks of all of the immunogenic responses (humoral, B cell phenotype and T cell clusters) in spite of these different elements being differentially associated with the immune signatures subsequently described it appears on initial reading. Is there any way the authors can add more clarity to this summary of the whole cohort as it came across as contradictory to me as it currently reads.

The authors highlight the differences in cellular and antibody responses between the whole virus inactivated primed individuals and those with boosting with mRNA vaccine. This is a very interesting point, and the authors justify that both approaches might be required in differing situations. Could the authors suggest why these different vaccine technologies might be associated with these varying immune responses?

I appreciate that this is not the focus of this study, but as the authors highlight the key to understanding the importance of the differing humoral and cellular responses is correlation with clinical outcomes. It would be useful to know if there is clinical follow up of this cohort and whether reinfections were monitored for. This is acknowledged in the discussion, but should perhaps be brought in earlier.

In the discussion lines 317, 319 and 331 could the authors make this clearer that they are referring to SARS-CoV-2 specific MBC, not global B cells?

Author Rebuttal, first revision:

See Inserted PDF

Reviewers' Comments:

Reviewer #1:

Remarks to the Author:

The authors have addressed my questions and suggestions but there are still some minor aspects that need further editing.

The abstract needs further work to capture the key findings e.g. in particular there is no mention of the superiority of mRNA vaccines across a number of measured outcomes. Also priming with BBIP induced higher Abs than what?

Authors' response: We thank Referee #1 for the comments; we edited the abstract in lines 42-48 to emphasize the superiority of antibody responses after mRNA vaccination and to improve clarity about the IgA response to BBIP vaccination.

Line 277 It is a bit simplistic to describe these mBCs as representing an activated phenotype.

Authors' response: We thank Referee #1 for pointing this out; we edited the corresponding text in lines 287-294 to better explain the phenotypes of the spike-binding mBCs.

Line 342-3 It's not clear why dengue is mentioned here and how it relates to the conclusions being drawn.

Authors' response: We agree with Reviewer #1 and edited the sentence to improve clarity in lines 360-362.

Line 351-3 I don't follow the authors thinking with this speculation. What cytokines specifically are they talking about and what proof do they have?

Authors' response: We have expanded a little on that and explain that we just speculate cytokines to be involved based on previous studies cited here (Lines 371-372).

Line 357-8 The T cells expressing high CD27 and CD45RA are therefore naïve T cells so this should be stated

Authors' response: We updated the discussion to mention that these markers are associated with naïve T cells on line 377.

Lines 364-5 Stating they “are involved in the immune response against viruses and other pathogens” is a very vague description of the role of DN cells and gives no indication of what they do. Given 2 CD4-/CD8- clusters are expanded they warrant further discussion.

Authors’ response: We welcome this suggestion and edited the corresponding text in lines 383-390 to better explain the role of DN T cells in immune responses.

Line 373 Maybe best to say “better protection” rather than “reliable protection”.

Authors’ response: We agree and changed it.

Line 443 "collected at T3" not "collected on T3"

Authors’ response: We thank the reviewer for the correction and edited the text accordingly.

Reviewer #3:

Remarks to the Author:

The authors elegantly study humoral and cellular responses to 16 homologous and heterologous prime boost vaccination schedules for SARS-CoV-2 encompassing a variety of vaccine technologies. This is a very useful study reflecting world experiencing in response to COVID-19 which has required adaptation of vaccine schedules based on availability.

Methodically characterising the immunological landscape across multiple vaccine combinations is highly informative for future vaccine design as previous vaccine studies have covered only limited combinations and comparisons of results are problematic.

The authors combine data on antibody responses, antigen specific B cell responses, T cell functional responses to SARS-CoV-2 peptides and global T cell subset frequencies to define four immune signatures associated with the varying SARS-CoV-2 vaccine combinations.

Appropriate methods and analysis have been used and the data is well presented, with a coherent description of the complex analysis of multiple subgroups. The discussion appropriately addresses the limitations of the study. As the authors conclude, this study has implications for considering which might be the most appropriate vaccine technologies and schedules for other

pathogens. Subsequent correlation of these results with protection from infection is the key to harnessing these results.

Authors' response: We thank the reviewer for the positive feedback and the astute summary of our work.

Suggestions and clarifications:

I have some comments regarding the clarity of the report and have highlighted points at which I think additional information would strengthen the manuscript. There is a wealth of data to display and at times it is difficult to ascertain the most pertinent results, especially with the number of extended figures. In specific instances detailed below I felt that key results are not reported in the text / figures at an early enough point and come much later than I was expecting.

Line 79: Novovax is mentioned in the introduction as an example of a recombinant spike nanoparticle vaccine. Whilst it is useful to highlight the other technologies available it should perhaps be made clearer that this is not a vaccine technology being assessed in the current study.

Authors' response: We thank Referee #3; we agree and we made this point clearer in the introduction section on lines 81-82.

Study design section: Whilst a broad range of ages are covered in this cohort, within the subgroups receiving the different vaccine regimens it is evident that there is some disparity. The mean age of some of the groups is higher than others, at the most extreme 54.7 for Sput26 followed by BBIBP, vs 26.2 for BBIBP followed by AZ, a difference of almost 30 years – are these significant difference? Given the age related changes in the immune response, including those observed to SARS-CoV-2 vaccination (Collier et al., Nature 2021; Ferrerira et al., medrxiv 2022) can the authors comment on this / address this potential limitation? This is especially relevant given the timing of this study in relation to recommendations about use of the AZ vaccine with concerns regarding thrombotic events – it would be useful to comment if any age related restrictions were imposed on the use of AZ vaccines at the time of this study in Argentina.

Authors' response: We thank the reviewer for pointing out the impact of age as a potential confounder in vaccine responses. In our study, the median age indeed differed slightly between some of the vaccine groups. All the PBMCs in this cohort were collected from adults older than 18 and under the age of 60 (mean age 41.8, standard deviation 15.6), reducing the likelihood of impaired immunogenicity due to

high age as shown in Collier et al. mainly for individuals above the age of 80 (Fig. 1a) and Ferreira et al. for individuals older than 70 years, but potentially not eliminating all age-related confounding factors between vaccination regimens.

Nevertheless we agree that some of the antibody titer analyses included patients which were indeed over 60. As you can see in the plots below, there is indeed a trend towards reduced antibody production in older volunteers. Nevertheless, a direct comparison of antibody titres in over and under 60 year old donors across all vaccine groups showed no significant difference in anti-S-RBD IgG titres. The Collier reference has been added to the revised version of our manuscript and age has been discussed as a potential limitation on lines 418-421 and the age range of the volunteers has been added for each analysis to the materials and methods section.

Regarding the correlation of adverse events and age: Most importantly, we did not observe any life-threatening adverse events across all individuals in our cohort. Moreover, according to the last Vaccine Safety Report issued by the National Ministry of Health (November 2022) the number and ratio of Severe Events Supposedly Attributable to Vaccination and Immunizations (ESAVIs in spanish) was extremely low (<https://bancos.salud.gob.ar/sites/default/files/2022-12/informe-19-noviembre-2022.pdf>, pages 6 and 7, in spanish).

At the time of the study there were no age related restrictions in place for the AZ vaccine in Argentina. AZ and Sput 26 were the first vaccines to be authorized in the country at the beginning of 2021. Thus, some groups in our study that received either AZD or Sput 26 as Dose 1 have a higher mean age because these two vaccines were the first applied for Dose 1 in Argentina, primarily to people ranging the mean age showed in Figure 1, according to vaccination priority strategy fixed by the Health Ministry of Argentina. Other vaccines used in this study were authorized later (BBIBP, AD5 and mRNA1273) or were available later (Sput 5), so they were received by younger people during this study.

Taken together, a) the single-cell analysis did not include volunteers over 60, b) there is only a minor influence of age on the antibody titers and c) we did not observe enough severe adverse events to study toxicity in the context of ageing.

The serological results are a key finding in terms of potential correlates of protection, initially only feature in the extended figures until comparisons with cellular components are analysed. Is it feasible to move some of these into the main figures? I note the addition of Figure 3e following the comments from Reviewer 1, and this is very important, however, I am not sure about its placement. I feel it would sit better with the data in the 'strength of antibody response varies between vaccines' section and had in fact made a note of its absence here, only to discover it was then addressed in the B cell phenotyping section. If at all possible I think it would be useful to move this earlier in the results.

Author's response: We thank the reviewer for the suggestion and moved the antibody results (Ext. data 2) to the main figures (Fig. 2).

Line 132 should this read anti-S RBD IgG and anti-S RBD IgA as mentioned in the preceding sentence?

Authors' response: We have corrected line 137 accordingly.

Lines 133-135 Why are only IgG and nAb results (Extended data Fig 2a,b) and not IgA featured here? What about the longitudinal IgA response following vaccination?

Authors' response: IgA measurements were not part of the original analysis and therefore not part of the originally submitted manuscript. Reviewer #2 suggested to analyze IgA for the revision. Due to sampling limitations at this point, we analyzed anti-S-RBD IgA across the 349 PBMC donors at T3 to correlate with our single-cell analysis. The data were then presented in the revised version of the manuscript (**Figure 2d, 4f, 7a-d and Extended Figure 3c,d,f**). The longitudinal analysis of antibody titers was performed independent of the immunophenotyping and restricted to IgG and NAb. The materials and methods section was updated accordingly.

Were they detectable in all and did they increase as with IgG? Additionally, there is no statement of the number of individuals who achieved detectable titres or whether a boosting effect was observed as seen with IgG and nAb (extended data Fig 3 a,b). Could these results be detailed in the text even if editorial restrictions mean it is not possible to show all of these data?

Authors' response: Excellent idea; to report the number of individuals with detectable IgA titres at T3, we added Fig. 4f and updated the results on lines 203-206.

Lines 146-149: The highest median values of IgG to RBD and nAb are observed with the homologous mRNA vaccine schedule, not with a heterologous schedule, but there is limited fold change as demonstrated by the high values already present at T1 after priming with this schedule. As these data are currently in the extended figures as opposed to the main body I think it may be worth explicitly stating this finding or making this text a little clearer.

Authors' response: We thank Referee #3; we agree and we made this point clearer on lines 143-147.

Line 153 Ref 37 – I think that other references may be more appropriate here with more B cell focus than this review.

Authors' response: We agree with the reviewer and we have added new references on line 159.

Line 156 – How are the authors defining spike specific memory B cells (which markers are used to define as being a MBC)? I could not see this detailed in the methods or the figure legend. Is it the same gating strategy as detailed in the supplementary data of the reference? Are the authors able to show example plots of the spike probe staining and did they use decoy probes or dual staining to help demonstrate the specificity of the staining?

Authors' response: We apologize for the lack of clarity; Spike-binding mBCs were defined using FlowSOM clustering as non-naive (IgD+IgM+ double positive naive B cells were excluded) and positive for SARS-CoV-2 spike multimers as described in Figure 3a, Extended Fig. 3a., Goel et al., Science, 2021 and Goel et al., Science Immunology, 2021. For clarification of this reviewer question, we show below a representative FACS plot of the gating strategy.

Neither decoy probes nor additional multimers were used for the analysis of spike-binding mBCs in our cohort of Argentine volunteers. Instead, we relied on fluorochrome-coupled streptavidin alone to control for cytophilic binding and false positives. Also, samples from unvaccinated and unexposed individuals were found to be negative. We have clarified this point in the materials and methods section on lines 522-524. Additional confidence in this assay was provided by the fact that mRNA-boosted individuals showed consistently higher spike-specific mBC frequencies compared to the other groups. In addition, spike-specific mBC frequencies correlated with the antibody response (Fig. 3e). If requested, we could provide this exemplary plot also as part of an extended figure.

Lines 167-171 I am not entirely sure what the authors are suggesting here – the spike MBC % correlates with serological results (as previously demonstrated in the literature), ‘this could serve as an indicator for the systemic antibody response in certain cases’. I initially read this to mean the authors are suggesting that cellular phenotyping could be used in lieu of serological analysis, which on a purely practical level I’m not sure would be of any use! Apologies if I have misunderstood this statement.

Authors’ response: We agree that this was misleading and therefore deleted the sentence.

I am concerned that the T cell ELISpot responses are somewhat overstated. Could the authors state the frequency of individuals with IFN- γ responses above the threshold at T1 and T3 shown in Fig. 5d earlier in the results as this seemed notably absent during the initial reading of the T cell responses section? Figure 4a and c demonstrate that a proportion of individuals (what seems a high proportion in some of the subgroups) do not have responses above the threshold that the authors have defined for detection, and the subsequent comparison of these results to show a significant increase in the

response when there is no above threshold response for a proportion of the individuals analysed feels a little misleading. One option is to demonstrate those who go from undetectable to detectable responses perhaps?

Authors' response: We added the number of volunteers with detectably IFN-g response at T1 and T3 to the new Fig. 6d in the revised version.

Additionally, the fold increase observed in the AZD/mRNA-1273 group seems to be largely driven by one extreme outlier – whilst I appreciate this is of course real world data it is important not to overstate this finding, and the BBICP primed groups seem much more convincing.

Authors' response: We agree with the reviewer that this group contains an outlier, however we calculated the statistical significance of the fold change with the Wilcoxon Rank Sum test, a non-parametric test that does not assume normality and is robust to outliers. Nonetheless, we agree and reworded to focus on the BBIBP-primed individuals in the results section on lines 224-226.

Line 229 – were any responses observed nucleocapsid peptides in the solely spike vaccinated individuals? 'Strong' seems an inappropriate word to use in this sentence, and it should be clarified if there were any responses to N protein observed in response to spike vaccination – as the authors state this would not be expected.

Authors' response: We thank the reviewer for the question and removed the word "strong". There was a response to N protein in a small proportion of spike-vaccinated individuals that was comparable to the one observed in unexposed individuals in Le Bert et al., Nature, 2020. We added a sentence to the manuscript on lines 239-243 in order to state this finding.

Line 255-258: this summary of the findings from IFN- γ positive responders is somewhat confusing for the total cohort, given the subsequent description of the differing immune signatures. IFN- γ positive responders have hallmarks of all of the immunogenic responses (humoral, B cell phenotype and T cell clusters) in spite of these different elements being differentially associated with the immune signatures subsequently described it appears on initial reading. Is there any way the authors can add more clarity to this summary of the whole cohort as it came across as contradictory to me as it currently reads.

Authors' response: Indeed we describe the general correlations in Fig. 6c and Ext. Fig. 5g, which hold true across vaccine regimens. The vaccine regimen-dependent immune signatures are then driven by the proportion of participants positive or negative for IFN γ production. We clarified the manuscript at lines 267-270.

The authors highlight the differences in cellular and antibody responses between the whole virus inactivated primed individuals and those with boosting with mRNA vaccine. This is a very interesting point, and the authors justify that both approaches might be required in differing situations. Could the authors suggest why these different vaccine technologies might be associated with these varying immune responses?

Author's response: We thank the reviewer for this interesting question. Of course we can only speculate on the mechanisms that lead to differences in the immune response to different types of vaccines. We added a short discussion of these points on lines 396-403.

I appreciate that this is not the focus of this study, but as the authors highlight the key to understanding the importance of the differing humoral and cellular responses is correlation with clinical outcomes. It would be useful to know if there is clinical follow up of this cohort and whether reinfections were monitored for. This is acknowledged in the discussion, but should perhaps be brought in earlier.

Authors' response: To make this clearer earlier in the manuscript, we added a sentence on lines 124-126. Analysis of effectiveness of the different combinations of vaccines was out of scope of our trial and not systematically studied. The only reliable data we have is that until July 2022 there were no deaths reported within the cohort.

In the discussion lines 317, 319 and 331 could the authors make this clearer that they are referring to SARS-CoV-2 specific MBC, not global B cells?

Authors' response: We had a careful read and fixed these items in the revised version of the manuscript.

Decision Letter, second revision:

25th Feb 2023

Dear Burkhard,

Thank you for submitting your revised manuscript "High-dimensional analysis of sixteen SARS-CoV-2 vaccine combinations reveals lymphocyte signatures correlating with immunogenicity" (NI-A34819B). Thank you for providing the revised manuscript. We are happy in principle to publish it in Nature Immunology, pending minor revisions to comply with our editorial and formatting guidelines.

We will now perform detailed checks on your paper and will send you a checklist detailing our editorial and formatting requirements in about a week. Please do not upload the final materials and make any revisions until you receive this additional information from us.

If you had not uploaded a Word file for the current version of the manuscript, we will need one before beginning the editing process; please email that to immunology@us.nature.com at your earliest convenience.

Thank you again for your interest in Nature Immunology. Please do not hesitate to contact me if you have any questions.

Kind regards,

Laurie

Laurie A. Dempsey, Ph.D.
Senior Editor
Nature Immunology
l.dempsey@us.nature.com
ORCID: 0000-0002-3304-796X

Final Decision Letter:

Dear Burkhard,

I am delighted to accept your manuscript entitled "High-dimensional analysis of sixteen SARS-CoV-2 vaccine combinations reveals lymphocyte signatures correlating with immunogenicity" for publication in an upcoming issue of Nature Immunology.

Over the next few weeks, your paper will be copyedited to ensure that it conforms to Nature Immunology style. Once your paper is typeset, you will receive an email with a link to choose the appropriate publishing options for your paper and our Author Services team will be in touch regarding any additional information that may be required.

Please note that *Nature Immunology* is a Transformative Journal (TJ). Authors may publish their research with us through the traditional subscription access route or make their paper immediately open access through payment of an article-processing charge (APC). Authors will not be required to make a final decision about access to their article until it has been accepted. [Find out more about Transformative Journals](https://www.springernature.com/gp/open-research/transformative-journals).

Your paper will be published online soon after we receive your corrections and will appear in print in the next available issue. Content is published online weekly on Mondays and Thursdays, and the embargo is set at 16:00 London time (GMT)/11:00 am US Eastern time (EST) on the day of publication. Now is the time to inform your Public Relations or Press Office about your paper, as they might be interested in promoting its publication. This will allow them time to prepare an accurate and satisfactory press release. Include your manuscript tracking number (NI-A34819C) and the name of the journal, which they will need when they contact our office.

About one week before your paper is published online, we shall be distributing a press release to news organizations worldwide, which may very well include details of your work. We are happy for your institution or funding agency to prepare its own press release, but it must mention the embargo date and Nature Immunology. Our Press Office will contact you closer to the time of publication, but if you or your Press Office have any enquiries in the meantime, please contact press@nature.com.

Also, if you have any spectacular or outstanding figures or graphics associated with your manuscript - though not necessarily included with your submission - we'd be delighted to consider them as candidates for our cover. Simply send an electronic version (accompanied by a hard copy) to us with a possible cover caption enclosed.

If you have not already done so, we strongly recommend that you upload the step-by-step protocols used in this manuscript to the Protocol Exchange. Protocol Exchange is an open online resource that allows researchers to share their detailed experimental know-how. All uploaded protocols are made freely available, assigned DOIs for ease of citation and fully searchable through nature.com. Protocols can be linked to any publications in which they are used and will be linked to from your article. You can also establish a dedicated page to collect all your lab Protocols. By uploading your Protocols to Protocol Exchange, you are enabling researchers to more readily reproduce or adapt the methodology you use, as well as increasing the visibility of your protocols and papers. Upload your Protocols at www.nature.com/protocolexchange/. Further information can be found at www.nature.com/protocolexchange/about .

Please note that we encourage the authors to self-archive their manuscript (the accepted version before copy editing) in their institutional repository, and in their funders' archives, six months after publication. Nature Portfolio recognizes the efforts of funding bodies to increase access of the research they fund, and strongly encourages authors to participate in such efforts. For information about our editorial policy, including license agreement and author copyright, please visit www.nature.com/ni/about/ed_policies/index.html

Kind regards,

Laurie

Laurie A. Dempsey, Ph.D.
Senior Editor
Nature Immunology

l.dempsey@us.nature.com
ORCID: 0000-0002-3304-796X